computational mathematics/mathematical modelling/applied mathematics

uncertainty quantification, transitional Markov chain Monte Carlo, inverse problem, epidemic modelling, Bayesian model selection, SIR model

**Author for correspondence:**
Anastasios Matzavinos
e-mail: matzavinos@brown.edu

# Data-driven prediction and origin identification of epidemics in population networks

Karen Larson[1], Georgios Arampatzis[3,4], Clark Bowman[5], Zhizhong Chen[2], Panagiotis Hadjidoukas[3], Costas Papadimitriou[6], Petros Koumoutsakos[3] and Anastasios Matzavinos[1]

[1]Division of Applied Mathematics, and [2]Department of Physics, Brown University, Providence, RI 02912, USA
[3]Computational Science and Engineering Laboratory, ETH Zürich, CH-8092, Switzerland
[4]Collegium Helveticum, CH-8092 Zürich, Switzerland
[5]Department of Mathematics and Statistics, Hamilton College, Clinton, NY 13323, USA
[6]Department of Mechanical Engineering, University of Thessaly, GR-38334 Volos, Greece

CP, 0000-0002-9792-0481; PK, 0000-0001-8337-2122; AM, 0000-0003-0491-7329

Effective intervention strategies for epidemics rely on the identification of their origin and on the robustness of the predictions made by network disease models. We introduce a Bayesian uncertainty quantification framework to infer model parameters for a disease spreading on a network of communities from limited, noisy observations; the state-of-the-art computational framework compensates for the model complexity by exploiting massively parallel computing architectures. Using noisy, synthetic data, we show the potential of the approach to perform robust model fitting and additionally demonstrate that we can effectively identify the disease origin via Bayesian model selection. As disease-related data are increasingly available, the proposed framework has broad practical relevance for the prediction and management of epidemics.

## 1. Introduction

Robust prediction of the spread of an epidemic is critical to monitoring and halting its progress. The reliability of these predictions, which have high clinical and societal significance, hinges on the underlying mathematical models which quantify the spread and virulence of diseases. Several models have been proposed for predicting the spread of epidemics in real-world populations, allowing for the

development of strategies for effectively managing disease spread via organized intervention. Perhaps the most well-known approach is Kermack & McKendrick's compartmental SIR model (and its extensions, such as SIRS and SEIR), a differential equation model which divides populations into groups corresponding to their relation with the disease (e.g. susceptible or recovered), which is widely studied due to its simplicity and predictiveness for several common diseases [1,2]. More recent work has also incorporated the topological aspect of network structure by modelling explicit (or random) population networks through which diseases propagate [3–6], working towards a more holistic view of disease modelling which can include aspects such as demography, land use and climate change [7]. The predictions made by these mathematical models have been used to study a diverse set of historical and modern epidemics, including HIV [8,9], malaria [10–12], polio [13] and tuberculosis [14], by using a wide range of data assimilation techniques [15,16].

Many aspects of these models have also been analysed from a more abstract mathematical perspective. Local bifurcation analysis has been performed on the man–environment–man and SIR models [17,18], while other work has used Lyapunov functions to determine endemic equilibria for SIRS and SEIR models [19,20] or has considered a mean-field approach [6,21–23]. Recent work has addressed how the network structure influences the spread of disease via the initial conditions and network topologies [4,5] and the ways in which epidemics spread on random networks [3]. Analytical results have also been obtained for the case of two competing (or promoting) diseases on a network [18,24]. Moreover, many of the models in question have been used to design intervention policies or allocate vaccines via optimal control [20,25] or randomized interventions [26].

In this work, we introduce data-driven reverse engineering of models for the spread of an epidemic through a population network. The model structure and parameters are inferred from noisy observations using a Bayesian framework for uncertainty quantification (UQ); Bayesian inference enables robust predictions and the rational selection of the best among competing models using data-based evidence. At the same time, Bayesian inference involves sampling of the (potentially high-dimensional) parameter space, requiring repeated evaluations of the forward model. As such, in cases where the forward model is complex and computationally intensive (e.g. a large network with intricate connectivity), the Bayesian approach may be prohibitively expensive. Yet the Bayesian setting is of considerable practical interest given its potential applications to real-world data collection: efficient parameter estimation would enable calibration of the models with actual observations, and models could be compared based on their degree of fit.

In what follows, we apply our Bayesian UQ framework $\Pi$4U to an extension of the SIR model to graphs. $\Pi$4U is an efficient parallel implementation of the transitional Markov chain Monte Carlo (TMCMC) algorithm, which offsets the complexity of UQ approaches by making use of modern parallel computation to run many copies of the model simultaneously [27–29]. Using noisy, simulated data, we show that our method is able to efficiently estimate values of the model parameters and their underlying uncertainties. The $\Pi$4U framework is also convenient for Bayesian model selection, which we use to identify the origin of the epidemic (e.g. [30]) by considering each possible starting location as a separate model. Bayesian approaches thus show significant potential to aid in real-world epidemic modelling and mitigation.

# 2. SIR model

The SIR epidemic model decomposes a population into three eponymous groups: hosts who are *susceptible* to the disease, hosts who are infected and contagious (the *infective* group), and hosts who are neither susceptible nor infected, either via gained immunity from recovery or due to a vaccine, quarantine policies or disease-related death (the *removed* group).

## 2.1. Single population model

Let $S(t)$, $I(t)$ and $R(t)$ denote the size of the susceptible, infective and removed groups, respectively, as functions of a continuous time $t$. The SIR model is based on three main assumptions: first, since the timescale on which the disease evolves is assumed to be much shorter than the timescale on which the population may evolve via e.g. births or natural deaths, the population $Y$ is assumed constant, and so

$$S(t) + I(t) + R(t) = Y \qquad (2.1)$$

for all $t$ (note that individuals killed by the disease are considered part of the removed group). Second, members of the population are assumed to come into contact uniformly at random and at a constant rate $\beta$—this parameter governs the rate at which an infection can spread. Finally, the infective population recovers (or is otherwise removed from the infectives via e.g. death) at a constant rate $\gamma$. These

assumptions can thus be visualized as

$$S \xrightarrow{\beta} I \xrightarrow{\gamma} R, \tag{2.2}$$

yielding the following set of ordinary differential equations:

$$\frac{dS(t)}{dt} = -\beta IS, \quad \frac{dI(t)}{dt} = \beta IS - \gamma I \quad \text{and} \quad \frac{dR(t)}{dt} = \gamma I. \tag{2.3}$$

Namely, at a particular time $t$, $S(t)$ susceptibles and $I(t)$ infectives come into contact at a rate $\beta$, yielding $\beta IS$ transitions from susceptible to infective (implicitly assuming that contact with an infective immediately infects a susceptible—if this assumption is not desired, the chance of disease transfer can be incorporated in $\beta$). Meanwhile, $I(t)$ infectives are removed at a rate $\gamma$, yielding $\gamma I$ transitions from infective to removed.

## 2.2. Epidemic model on graphs

The SIR model is readily generalized to a directed graph with $N$ vertices, a mathematical construct which can be thought of as modelling a collection of $N$ distinct communities. Namely, let each node (community) be a distinct population whose dynamics evolve according to equation (2.3); the directed edges (connections between communities) are a convenient framework to dictate transfer between populations. Since each population itself has three groups (susceptible, infective, removed), three quantities are needed to describe movement. Here, we use $\lambda_{i,j}$, $\eta_{i,j}$ and $g_{i,j}$ to describe the rate of movement from node $i$ to node $j$ on the susceptible, infective and removed groups, respectively; identifying each transition rate as the weight of the edge connecting $i$ to $j$, these rates are naturally written as weighted adjacency matrices, here denoted $\Lambda$, $H$ and $G$. The SIR model on a network, now a system of $N$ models corresponding to each population $i$, can then be written as

$$\left.\begin{aligned}
\frac{dS_i(t)}{dt} &= -\beta I_i S_i + \sum_{j=1}^{N} \lambda_{j,i} S_j - \sum_{j=1}^{N} \lambda_{i,j} S_i, \\
\frac{dI_i(t)}{dt} &= \beta I_i S_i - \gamma I_i + \sum_{j=1}^{N} \eta_{j,i} I_j - \sum_{j=1}^{N} \eta_{i,j} I_i \\
\frac{dR_i(t)}{dt} &= \gamma I_i + \sum_{j=1}^{N} g_{j,i} R_j - \sum_{j=1}^{N} g_{i,j} R_i,
\end{aligned}\right\} \tag{2.4}$$

and

or more succinctly in matrix form:

$$\left.\begin{aligned}
\frac{dS}{dt} &= -\beta I \cdot S + \Lambda^T S - (\Lambda F) \cdot S, \\
\frac{dI}{dt} &= \beta I \cdot S - \gamma I + H^T I - (HF) \cdot I \\
\frac{dR}{dt} &= \gamma I + G^T R - (GF) \cdot R.
\end{aligned}\right\} \tag{2.5}$$

and

(Here, $F = (1, 1, \ldots, 1)^T$ is a vector of ones which simplifies the notation.) It should be emphasized that $S$, $I$ and $R$ are $N \times 1$ vectors whose $i$th element corresponds to the $i$th population. Note that if $\lambda_{i,j} = \eta_{i,j} = g_{i,j} = 0$ for all $i$, $j$, i.e. there is no movement between populations, each model reduces to the single population model (equation (2.3)).

# 3. Bayesian methodology

The network model described by equation (2.5) is a predictive model for tracking the spread of an epidemic through a population network. Here, we introduce a Bayesian approach to the inverse problem, i.e. reverse-engineering aspects of the model itself using observed outputs. In real-world scenarios, these observed outputs (e.g. the number of infected patients at a particular set of community health centres) are noisy due both to observational noise (e.g. not all infections are reported) and to model error—epidemic models are mathematical equations introduced to represent the real system, and so will not exactly predict the noise-free measurements. In particular, the proposed network model does not account for the intrinsic stochasticity of system parameters, such as the recovery rate, which could in reality vary based on a number of external factors including patient physiology, health centre availability, treatment options and more.

The inverse problem for epidemics is of considerable practical interest: accurate estimation of model parameters would allow for the identification of a disease's underlying characteristics (e.g. its infectivity $\beta$ or the host recovery rate $\gamma$). Moreover, by defining a class of models corresponding to the disease having originated in different communities, Bayesian model selection could be used to probabilistically determine the initial outbreak location [30–32], thereby aiding in identification and mitigation of the vector of infection. We note that stochastic optimization techniques such as CMA-ES [33] may also be used to infer optimal model parameters from noisy data [34]; however, such techniques neither enable robust predictions nor provide a framework for model selection as does the following Bayesian framework.

## 3.1. Bayesian uncertainty quantification

Denote as $\underline{\theta} \in \mathbb{R}^n$ the set of parameters corresponding to the model $M$ of interest. Here, the model $M$ is given by the SIR network model (equation (2.5)); its parameters include the infectivity $\beta$ and recovery rate $\gamma$. By evolving the system forward in time, we can generate deterministic predictions for the system at a future time $T$—for example, the number of infected individuals present at a certain subset of nodes.

We first consider the problem of parameter estimation: suppose we observe a subset of noisy predictions from this model and wish to estimate the parameters $\underline{\theta} \in \mathbb{R}^n$ which generated them. In particular, we will assume the observed data $\underline{D} \in \mathbb{R}^m$ obey the model-prediction equation

$$\underline{D} = \underline{g}(\underline{\theta}|M) + \underline{e}, \tag{3.1}$$

where $\underline{g}(\underline{\theta}|M) : \mathbb{R}^n \to \mathbb{R}^m$ denotes the deterministic mapping of parameters to outputs and $\underline{e}$ is an additive error term. The posterior distribution of the parameters given the data is then given by Bayes' theorem as

$$p(\underline{\theta}|\underline{D}, M) = \frac{p(\underline{D}|\underline{\theta}, M)\pi(\underline{\theta}|M)}{\rho(\underline{D}|M)} \tag{3.2}$$

in terms of the prior $\pi(\underline{\theta}|M)$, likelihood $p(\underline{D}|\underline{\theta}, M)$ and evidence $\rho(\underline{D}|M)$ of the model class, given by the multi-dimensional integral

$$\rho(\underline{D}|M) = \int_{\mathbb{R}^n} p(\underline{D}|\underline{\theta}, M)\pi(\underline{\theta}|M)d\underline{\theta}. \tag{3.3}$$

This scenario can be extended to the case where the model $M$ is one of many models in a parametrized class $\mathcal{M}$; the probability that the observed data were generated by any particular model $M_i$ is also given by Bayes' theorem:

$$Pr(M_i|\underline{D}) = \frac{\rho(\underline{D}|M_i)Pr(M_i)}{p(\underline{D}|\mathcal{M})}. \tag{3.4}$$

In particular, under the assumption of a uniform prior on models, $Pr(M_i|\underline{D})$ is directly proportional to the evidence $\rho(\underline{D}|M_i)$, and so model selection is 'free' when the evidence is already calculated for parameter estimation [35,36].

In order to calculate the likelihood $p(\underline{D}|\underline{\theta}, M)$ needed for equation (3.2), we need to postulate a probability model for the error term $\underline{e}$. Here, we assume the model error $\underline{e}$ is normally distributed with zero mean and covariance matrix $\Sigma$; the multivariate normal distribution maximizes entropy over the class of probability distributions on $\mathbb{R}^m$ with specified mean and covariance matrix [37]. Assuming that errors at different nodes are uncorrelated, the covariance matrix becomes $\Sigma = \sigma^2 I$, where $I$ is the $m \times m$ identity matrix.

If $\underline{e}$ is Gaussian, it follows that $\underline{D}$ is also Gaussian, and so the likelihood $p(\underline{D}|\underline{\theta}, M)$ of the observed data is given as

$$p(\underline{D}|\underline{\theta}, M) = \frac{|\Sigma(\theta)|^{-1/2}}{(2\pi)^{m/2}}\exp\left[-\frac{1}{2}J(\underline{\theta}, \underline{D}|M)\right], \tag{3.5}$$

where

$$J(\underline{\theta}, \underline{D}|M) = [\underline{D} - \underline{g}(\underline{\theta}|M)]^T \Sigma^{-1}(\theta)[\underline{D} - \underline{g}(\underline{\theta}|M)] \tag{3.6}$$

is the weighted measure of fit between the model predictions and the measured data, $|\cdot|$ denotes determinant, and the parameter set $\underline{\theta}$ is augmented to include parameters that are involved in the structure of the covariance matrix $\Sigma$ (here, the noise level $\sigma$).

The main computational barrier in calculating the posterior distribution of parameters given by equation (3.2) is the complex forward problem $g$ (the epidemic network model) which appears in the fitness $J(\underline{\theta}, \underline{D}|M)$.

**Algorithm 1.** TMCMC

1: procedure TMCMC [28]

2: BEGIN, SET $j = 0, q_0 = 0$

3: **Generate** $\{\underline{\theta}_{0,k}, k = 1, \ldots, N_0\}$ from prior $f_0(\underline{\theta}) = \pi(\underline{\theta}|M)$ and compute likelihood $p(\underline{D}|\underline{\theta}_{0,k}, M)$
   for each sample.

4: **while** $q_{j+1} \leq 1$ **do:**

5:   **Analyse** samples $\{\underline{\theta}_{j,k}, k = 1, \ldots, N_j\}$ to determine $q_{j+1}$, weights $\overline{w}(\underline{\theta}_{j,k})$, covariance $\Sigma_j$,

6:    and estimator $S_j$ of $\mathbb{E}[w(\underline{\theta}_{j,k})]$.

7:   **Resample** based on samples available in stage $j$ in order to generate samples for stage $j + 1$

8:    and compute likelihood $p(\underline{D}|\underline{\theta}_{j+1,k}, M)$ for each.

9:   **if** $q_{j+1} > 1$ **then**

10:       BREAK,

11:   **else**

12:       $j = j + 1$

13:   **end**

14: **end loop**

15: **end procedure**

The $\Pi$4U software [28] has two advantages in this respect: first, it approximately samples the posterior via transitional Markov chain Monte Carlo [38], described below, which is massively parallelizable; and second, it leverages an efficient parallel architecture for task sharing (see Appendix).

## 3.2. Transitional Markov chain Monte Carlo

The TMCMC algorithm used by $\Pi$4U functions by transitioning to the target distribution (the posterior $p(\underline{\theta}|\underline{D}, M)$) from the prior $\pi(\underline{\theta}|M)$. To accomplish this, a series of intermediate distribution are constructed iteratively:

$$f_j(\underline{\theta}) \sim [p(\underline{D}|\underline{\theta}, M)]^{q_j} \cdot \pi(\underline{\theta}|M), \quad j = 0, \ldots, \lambda$$
$$0 = q_0 < q_1 < \ldots < q_\lambda = 1. \tag{3.7}$$

The explicit algorithm, shown as algorithm 1, begins by taking $N_0$ samples $\underline{\theta}_{0,k}$ from the prior distribution $f_0(\underline{\theta}) = \pi(\underline{\theta}|M)$. For each stage $j$ of the algorithm, the current samples are used to compute the plausibility weights $w(\underline{\theta}_{j,k})$ as

$$w(\underline{\theta}_{j,k}) = \frac{f_{j+1}(\underline{\theta}_{j,k})}{f_j(\underline{\theta}_{j,k})} = [p(\underline{D}|\underline{\theta}_{j,k}, M)]^{q_{j+1}-q_j}.$$

Recent literature suggests that $q_{j+1}$, which determines how smoothly the intermediate distributions transition to the posterior, should be taken to make the covariance of the plausibility weights at stage $j$ smaller than a tolerance covariance value, often 1.0 [28,38].

Next, the algorithm calculates the average $S_j$ of the plausibility weights, the normalized plausibility weights $\overline{w}(\underline{\theta}_{j,k})$ and the scaled covariance $\Sigma_j$ of the samples $\underline{\theta}_{j,k}$, which is used to produce the next generation of samples $\underline{\theta}_{j+1,k}$:

$$\left. \begin{aligned} S_j &= \frac{1}{N_j} \sum_{k=1}^{N_j} w(\underline{\theta}_{j,k}), \\ \overline{w}(\underline{\theta}_{j,k}) &= \frac{w(\underline{\theta}_{j,k})}{\sum\limits_{k=1}^{N_j} w(\underline{\theta}_{j,k})} = \frac{w(\underline{\theta}_{j,k})}{N_j S_j} \end{aligned} \right\} \tag{3.8}$$

and
$$\Sigma_j = b^2 \sum_{k=1}^{N_j} \overline{w}(\underline{\theta}_{j,k})[\underline{\theta}_{j,k} - \underline{\mu}_j][\underline{\theta}_{j,k} - \underline{\mu}_j]^T.$$

$\Sigma_j$ is calculated using the sample mean $\underline{\mu}_j$ of the samples and a scaling factor $b$, usually 0.2 [28,38].

The algorithm then generates $N_{j+1}$ samples $\hat{\underline{\theta}}_{j+1,k}$ by randomly selecting from the previous generations of samples $\{\underline{\theta}_{j,k}\}$ such that $\hat{\underline{\theta}}_{j+1,\ell} = \underline{\theta}_{j,k}$ with probability $\overline{w}(\underline{\theta}_{j,k})$. These samples are selected independently at random, so any parameter can be selected multiple times—call $n_{j+1,k}$ the number of times $\underline{\theta}_{j,k}$ is selected. Each unique sample is used as the starting point of an independent Markov chain of length $n_{j+1,k}$ generated using the Metropolis algorithm with target distribution $f_j$ and a Gaussian proposal distribution with covariance $\Sigma_j$ centred at the current value.

Finally, the samples $\underline{\theta}_{j+1,k}$ are generated for the Markov chains, with $n_{j+1,k}$ samples drawn from the chain starting at $\underline{\theta}_{j,k}$, yielding $N_{j+1}$ total samples. Then the algorithm either moves forward to generation $j + 1$ or terminates if $q_{j+1} > 1$.

# 4. Results

In the following results, we apply our high-performance implementation of Bayesian UQ $\Pi$4U to a case study with simulated data from two example network structures. In each case, we fix particular values of the system parameters (infectivity $\beta$, recovery rate $\gamma$ and time of observation $T$) and use equation (2.5) to evolve the network forward in time via a fourth-order Runge–Kutta method. At the observation time $T$, the infective populations (and sometimes also the recovered populations) from a selected subset of communities, corrupted by additive Gaussian noise with noise level $\sigma$, are output as the noisy observed data. Namely, observed data $D_k$ are generated as

$$D_k = p_k + \sigma\epsilon_k,$$

where each deterministic population datum $p_k$ generated from the reference model is added to a zero-mean, unit-variance Gaussian $\varepsilon_k$ scaled by the noise level $\sigma$ to yield the observed noisy datum $D_k$. In order that the signal-to-noise ratio be high enough for meaningful estimation, we choose $\sigma$ to be a fraction $\sigma = 0.01\alpha$ (or sometimes $\sigma = 0.05\alpha$) of the average value $\alpha$ of all model outputs $p_k$.

We then use our method to approximately solve the inference problem by generating $10^4$ samples from the posterior distribution of the model parameters given these noisy outputs, checking the validity of our approach by comparing the resulting distributions to the known reference values. While we use synthetic, model-generated noisy data, we note that the framework is readily extended to the incorporation of real-world data.

For ease of comparison, we present numerical results in terms of the rescaled parameters ($\theta_\beta$, $\theta_\gamma$, $\theta_T$, $\theta_{\sigma/\alpha}$), given by $\theta_\beta = \beta/\beta_0$, i.e. the ratio between the estimated value and the true reference value. Accurate estimation thus results in scaled parameters close to 1. As the $\Pi$4U approach does not rely on the choice of a particular prior, we use a simple uniform distribution on $[0.01, 2] \times [0.5, 2] \times [0.02, 5] \times [0.005, 0.10]$ in this scaled parameter space. The uniform distribution, which maximizes entropy over a compact domain, functions as an agnostic prior: we assume as little knowledge as possible of the parameters (other than a generous feasible range) [37].

## 4.1. Network 1: the 20-barbell graph

We first consider a network with two distinct populations, each with many highly interacting sub-communities. The two populations mix via a single route, modelled by a single connecting edge. This 'barbell graph'—two complete 20-node graphs connected by a single edge—is illustrated in figure 1. In this case, we impose uniform transition rates between adjacent vertices of 0.02, 0.3 and 0.05 for the susceptible, infective and recovered populations, respectively. The infection begins at node 1 with the configuration $S_1(0) = 5$, $I_1(0) = 95$, $R_1(0) = 0$, and all other nodes are fully susceptible with configuration $S_i(0) = 100$, $I_i(0) = R_i(0) = 0$, $i \neq 1$.

We consider, in particular, the case of having information only from a limited subset of nodes; by placing the 'sensors' at different locations (i.e. observing different subsets of nodes), we can test how the sensor configuration influences the parameter estimation procedure and the corresponding uncertainties. In particular, we assume observations of both the infective and recovered populations at the sensor locations.

We consider three sensor configurations: in the first experiment, we place two sensors at nodes 3 and 12, which are members of the same complete subgraph as node 1, the origin of the epidemic (figure 1). In the second experiment, we gather data from both complete subgraphs by placing sensors at nodes 3 and 24. Finally, in the third experiment, we focus on nodes 24 and 27, which are part of the initially healthy complete subgraph. The disease evolves according to reference values $\beta = 0.02$ and $\gamma = 0.3$, while the

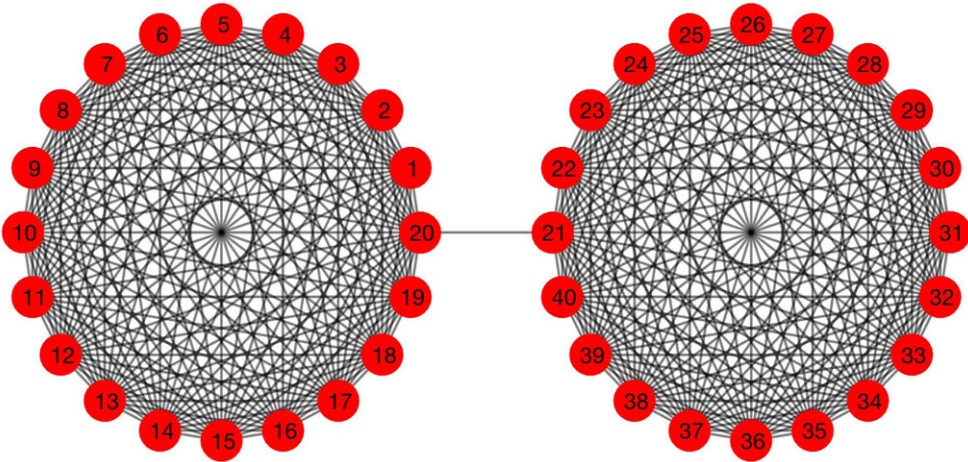

**Figure 1.** The 20-barbell graph. Two complete 20-node graphs are connected by a single edge.

observation time $T$ is chosen as $T = 3$, $T = 5$ and $T = 7$ in three separate cases, yielding nine total experiments (three sensor configurations each observed at three different times).

Results at the intermediate time $T = 5$ are summarized in the second block of table 1 and displayed in figure 2*a,b,c*. For all sensor configurations, reference values for $\beta$, $\gamma$ and $T$ are within two standard deviations of the estimated values, though the level of uncertainty varies significantly. In particular, estimated values were the most accurate and had the lowest uncertainty when using sensors at nodes 3 and 24, which were on opposite sides of the graph. Placing sensors at nodes 3 and 12, on the same side of the graph as the epidemic origin, yielded the least accurate estimates.

$\beta$ and $\gamma$ are positively correlated, i.e. similar outputs can be achieved by simultaneously raising both the infection rate and the recovery rate. Intuitively, a faster-spreading disease must be counteracted by quicker recovery in order for the dynamics to remain consistent. Similarly, both $\beta$ and $\gamma$ are negatively correlated with $T$; a more infectious disease or quicker recovery would increase the speed of the system dynamics, meaning similar outputs would be observed earlier.

The recovered noise standard deviation $\sigma/\alpha$ has significantly larger uncertainty than do the system parameters $\beta$, $\gamma$ and $T$. Despite this comparatively large uncertainty, the reference noise value $\sigma$ is recovered to within 2 s.d. for all three sensor configurations, though the posterior means $\theta_{\sigma/\alpha}$ consistently overestimate the true magnitude.

Figure 3 shows the deterministic populations of the susceptible, infected and recovered groups as a function of time for a selection of nodes involved in the experiments (since e.g. node 12 is identical to node 3, only one is shown). Nodes 21 and 27, both contained in the initially susceptible subgraph, reach peak infective population around time $t \approx 5$. Nodes on the side of the infection origin, conversely, achieve peak infective population at $t \approx 3$. The results of parameter estimation at the reference observation time $T = 5$ thus suggest that observing nodes around the time when the infective population peaks improves the accuracy of the recovered parameters. This effect can also be seen in figure 2 in the joint marginals of the system parameters $\beta$, $\gamma$ and $T$: nodes 3 and 12, which peak at $t \approx 3$, give much more smeared marginals in figure 2*a*, obtained using observations at time $T = 5$, than in figure 2*d*, corresponding to observations at $T = 3$; meanwhile nodes 24 and 27, which peak at $t \approx 5$, have sharper distributions in figure 2*c*, with observations taken at $T = 5$, than in figure 2*f*, which has observations at $T = 3$. The parameter distributions obtained for data from nodes 3 and 24, shown in figure 2*b* and figure 2*e*, are very similar in both cases, suggesting that placing one sensor in each subgraph leverages information from both timescales.

The parameter estimation results at $T = 7$, shown in the left column of figure 4, corroborate this conclusion. Though all nodes in the graph are well-past peak infective population at this time, using information on two different timescales (the two subgraphs) yields much sharper joint marginals (e.g. the joint distribution of $\beta$ and $T$ in figure 4*b* when compared with figure 4*a* and 4*c*).

Numerical values for $T = 3$ and $T = 7$ appear in the first and third block of table 1. In most cases, the parameters $\beta$, $\gamma$ and $T$ are recovered to within 1 s.d.; the notable exception is the configuration with sensors at nodes 3 and 12, which fails to recover $\beta$ within 2 s.d. when $T = 7$, by which time the disease has largely run its course in the left half of the graph. The experiment with sensors at nodes 3 and 24

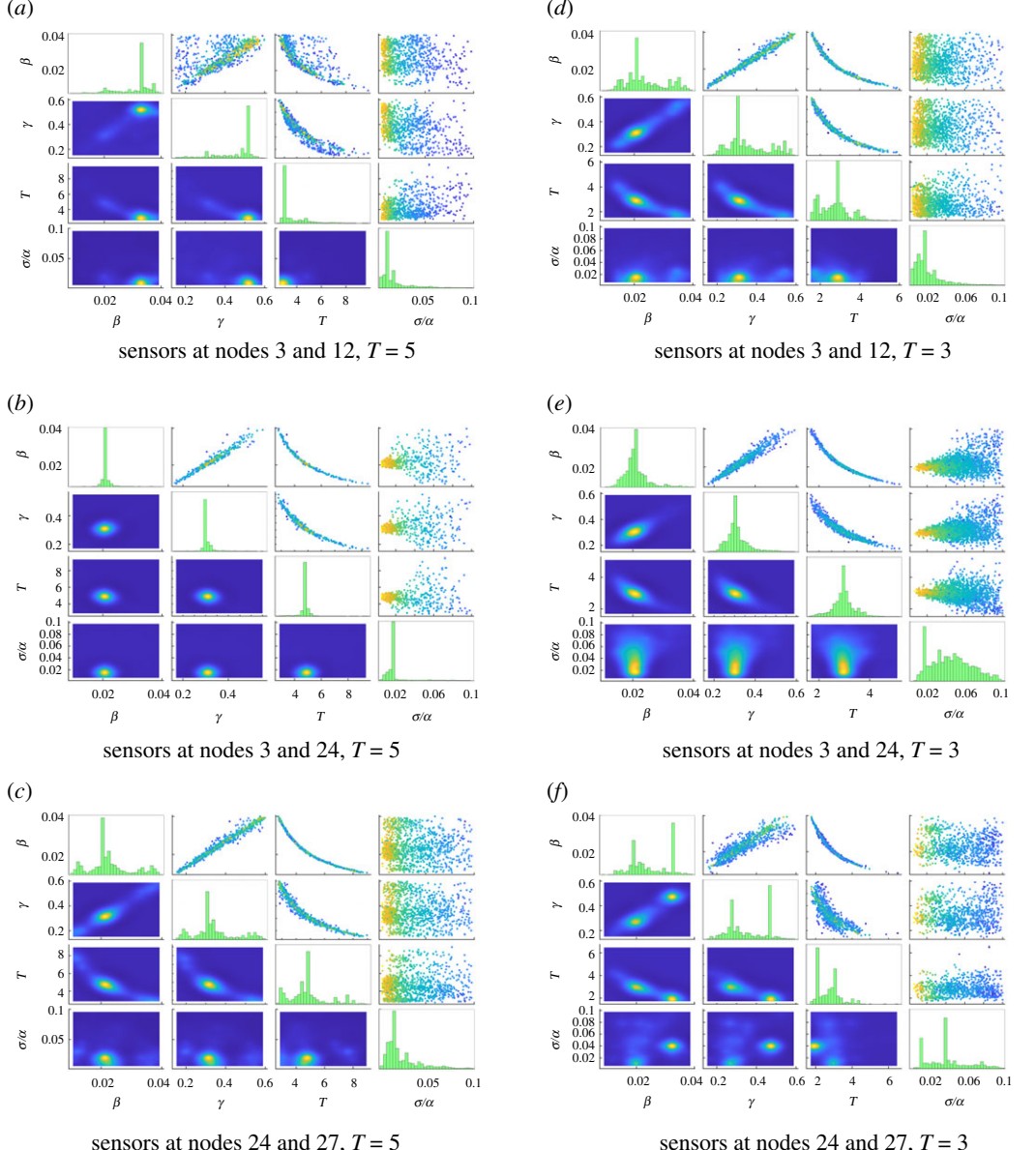

**Figure 2.** Parameter estimation results for the 20-barbell graph with infection rate $\beta = 0.02$, recovery rate $\gamma = 0.3$, noise level $\sigma = 0.01\alpha$ and time step $\Delta t = 0.005$. In each experiment, noisy data from two nodes were used to track the epidemic. For each pair of nodes, the experiment was run for time of observation $T = 5$ (a–c), $T = 3$ (d–f) and $T = 7$ (shown in figure 4). Histograms for each parameter are displayed along the main diagonal of the subfigure. Subfigures below the diagonal show the marginal joint density functions for each pair of parameters, while subfigures above the diagonal show the samples used in the final stage of TMCMC. Colours correspond to probabilities, with yellow likely and blue unlikely. (a) Sensors at nodes 3 and 12, $T = 5$, (b) sensors at nodes 3 and 24, $T = 5$, (c) sensors at nodes 24 and 27, $T = 5$, (d) sensors at nodes 3 and 12, $T = 3$, (e) sensors at nodes 3 and 24, $T = 3$, (f) sensors at nodes 24 and 27, $T = 3$.

continues to be both the overall most accurate (in terms of posterior means) and precise (in terms of uncertainties), highlighting the importance of sensor placement in leveraging information from different timescales.

To test the robustness of the parameter estimation to increased noise, we next reconsider the time $T = 5$ case with noise level $\sigma = 0.05\alpha$, five times the previously used $\sigma = 0.01\alpha$. Results appear in the right column of figure 4 and in the fourth block of table 1. Again, the experiment using simulated data from nodes 3 and 24 (figure 4e) recovers the parameters with comparatively lower uncertainty and greater accuracy. For all three sensor configurations, parameters are recovered to within one standard deviation, and so we conclude that the Bayesian UQ approach to SIR models on the 20-barbell graph has significant robustness to observational noise.

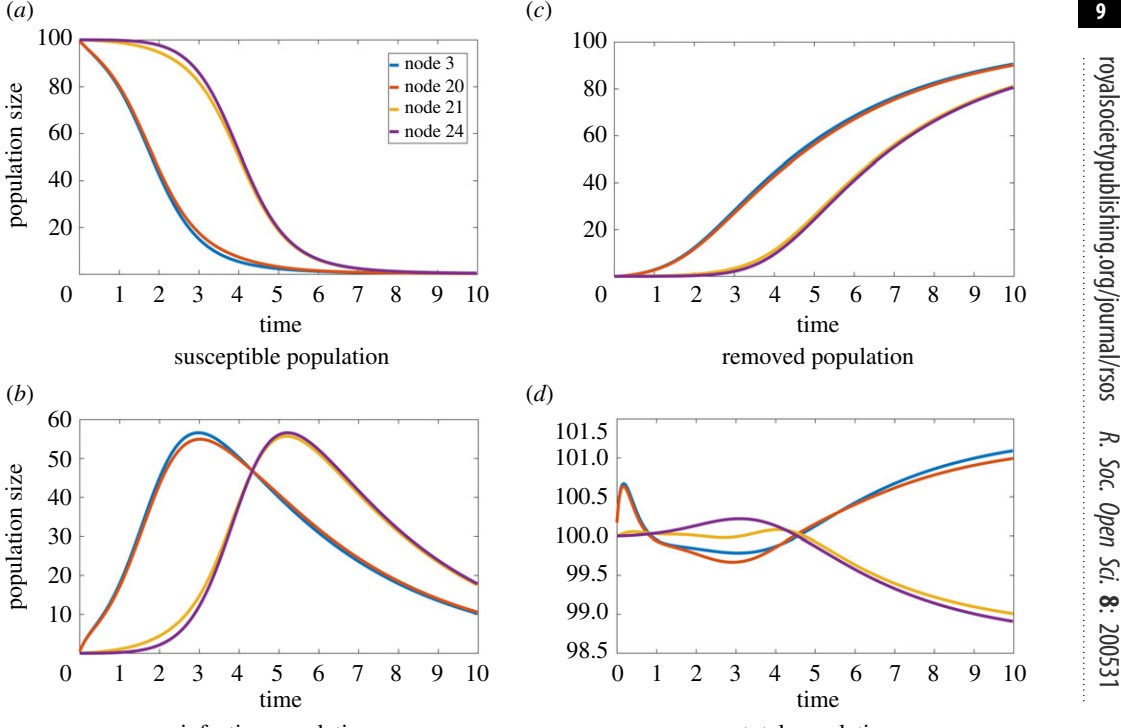

**Figure 3.** Time evolution of the susceptible, infective, recovered and total populations at nodes 3, 20, 21 and 24 of the 20-barbell graph. Nodes from different complete subgraphs have different trends and peak times. (*a*) Susceptible population, (*b*) infective population, (*c*) removed population, (*d*) total population.

**Table 1.** Numerical results for estimation of $\beta$, $\gamma$, $T$ and $\sigma/\alpha$ for the 20-barbell graph. Blocks of the table are different times of observation or noise levels. Parameters are reported in terms of scaled values $\theta$; accurate estimation thus results in values close to 1. Uncertainties, e.g. $u_\beta$, are the ratio of a parameter's standard deviation to its mean.

| data | node pair | $\theta_\beta$ | $u_\beta$ (%) | $\theta_\gamma$ | $u_\gamma$ (%) | $\theta_T$ | $u_T$ (%) | $\theta_{\sigma/\alpha}$ | $u_{\sigma/\alpha}$ (%) |
|---|---|---|---|---|---|---|---|---|---|
| $T = 3$, $\sigma = 0.01\alpha$ | 3 and 12 | 1.1889 | 27.89 | 1.1829 | 28.63 | 0.9064 | 25.89 | 2.08 | 70.51 |
| | 3 and 24 | 1.0435 | 17.83 | 1.0360 | 16.20 | 0.9870 | 13.89 | 4.53 | 46.94 |
| | 24 and 27 | 1.2369 | 27.08 | 1.1765 | 27.42 | 0.8925 | 22.99 | 4.07 | 53.71 |
| $T = 5$, $\sigma = 0.01\alpha$ | 3 and 12 | 1.5593 | 21.48 | 1.5235 | 21.17 | 0.6939 | 29.56 | 1.67 | 130.59 |
| | 3 and 24 | 1.0468 | 9.50 | 1.0225 | 8.46 | 0.9666 | 7.93 | 1.85 | 55.98 |
| | 24 and 27 | 1.1402 | 30.17 | 1.1303 | 30.37 | 0.9593 | 26.71 | 2.51 | 69.50 |
| $T = 7$, $\sigma = 0.01\alpha$ | 3 and 12 | 1.5858 | 17.56 | 1.5532 | 24.65 | 0.7032 | 36.88 | 1.33 | 76.57 |
| | 3 and 24 | 1.0906 | 21.41 | 1.0553 | 16.56 | 0.9581 | 16.94 | 2.24 | 67.16 |
| | 24 and 27 | 1.1675 | 27.95 | 1.1302 | 28.96 | 0.9315 | 21.72 | 2.33 | 67.11 |
| $T = 5$, $\sigma = 0.05\alpha$ | 3 and 12 | 1.0729 | 37.32 | 1.0956 | 36.02 | 1.0396 | 37.72 | 0.786 | 50.94 |
| | 3 and 24 | 1.1842 | 29.17 | 1.1151 | 26.41 | 0.9218 | 28.54 | 1.154 | 34.82 |
| | 24 and 27 | 1.2432 | 35.11 | 1.1968 | 35.50 | 0.9101 | 33.09 | 0.828 | 47.01 |

A final experiment tests the approach in the context of minor model misspecification. We perturb the network by introducing an extra edge between nodes 2 and 40; while observations are generated from a network which includes this edge, we perform parameter estimation using the original model (without the extra edge). Results for observing nodes 3 and 24 for the $T = 3$, $\sigma = 0.01\alpha$ case with two different perturbation strengths appear in figure 5 and table 2. When compared with the original case (second

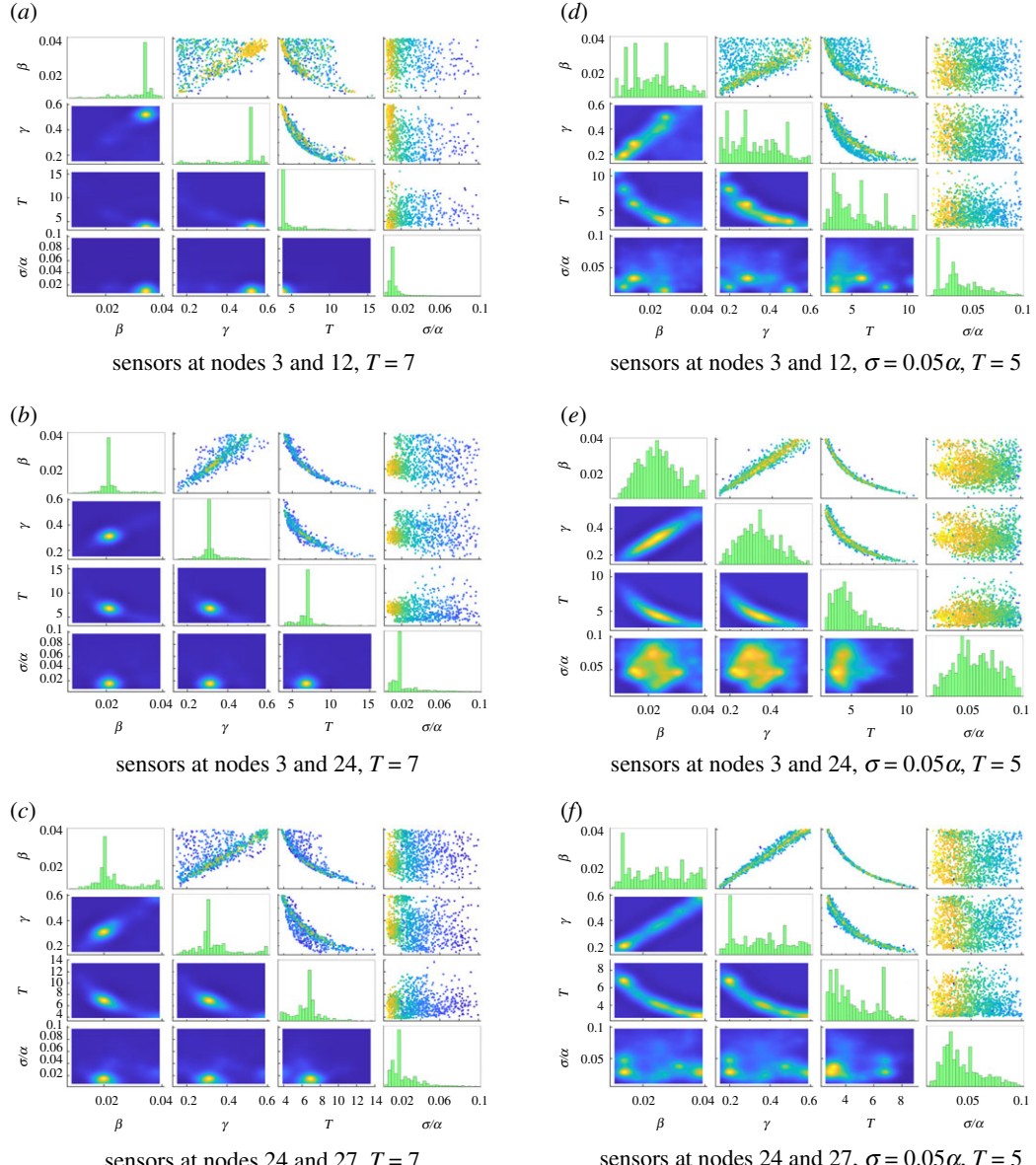

**Figure 4.** Parameter estimation results for the 20-barbell graph with infection rate $\beta = 0.02$, recovery rate $\gamma = 0.3$, time step $\Delta t = 0.005$ and noise level $\sigma = 0.01\alpha$ at time $T = 7$ (a–c) and with increased noise level $\sigma = 0.05\alpha$ at time $T = 5$ (d–f). See figure 2 for description of subfigures. (a) sensors at nodes 3 and 12, $T = 7$, (b) sensors at nodes 3 and 24, $T = 7$, (c) sensors at nodes 24 and 27, $T = 7$, (d) sensors at nodes 3 and 12, $\sigma = 0.05\alpha$, $T = 5$, (e) sensors at nodes 3 and 24, $\sigma = 0.05\alpha$, $T = 5$, (f) sensors at nodes 24 and 27, $\sigma = 0.05\alpha$, $T = 5$.

row of table 1 and figure 2e), the most notable change is the significant increase in the estimated noise level—disagreements between the original model (now misspecified) and the observed data are reconciled by assuming a much higher magnitude of noise. Nonetheless, estimation of system parameters ($\beta$, $\gamma$, $T$) for both perturbations is reasonably successful, with all reference values recovered to within two standard deviations, though uncertainties in estimation are roughly twice as large as without the perturbation.

## 4.2. Network 2: the three-group network

The second network considered is a 44-node graph comprising three large sub-networks with limited interaction (figure 6). Each sub-network has a distinct topological structure and set of non-uniform transition rates (explicit values appear in the appendix). We again consider three sensor configurations: a 7-node set (nodes 1–4, 20, 31 and 34), a 23-node set (nodes 1–7, 20–28 and 31–37) and a 35-node set (nodes 1–28 and 31–37), each in the presence of observational noise $\sigma = 0.01\alpha$. Parameter estimation

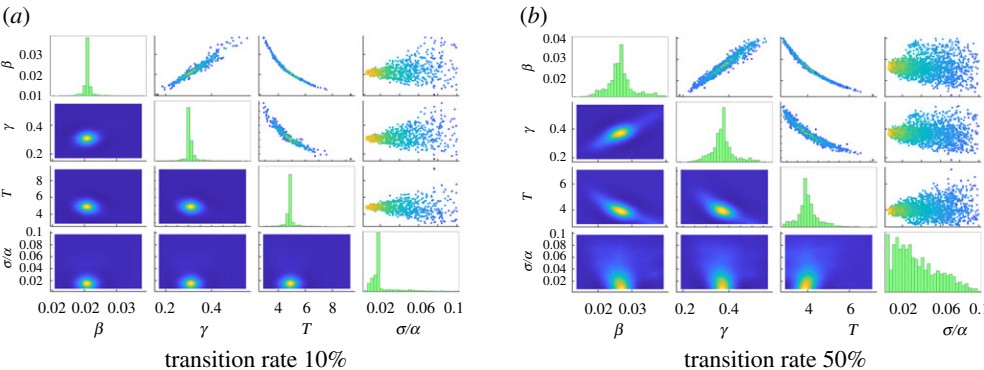

**Figure 5.** Parameter estimation results for the perturbed 20-barbell graph with infection rate $\beta = 0.02$, recovery rate $\gamma = 0.3$, time step $\Delta t = 0.005$ and noise level $\sigma = 0.01\alpha$ at time $T = 3$ using observations from nodes 3 and 24. Data are generated with an additional edge between nodes 2 and 40, which uses (a) 10% or (b) 50% of the transition rate of other edges; parameter estimation uses the original model. See figure 2 for description of subfigures.

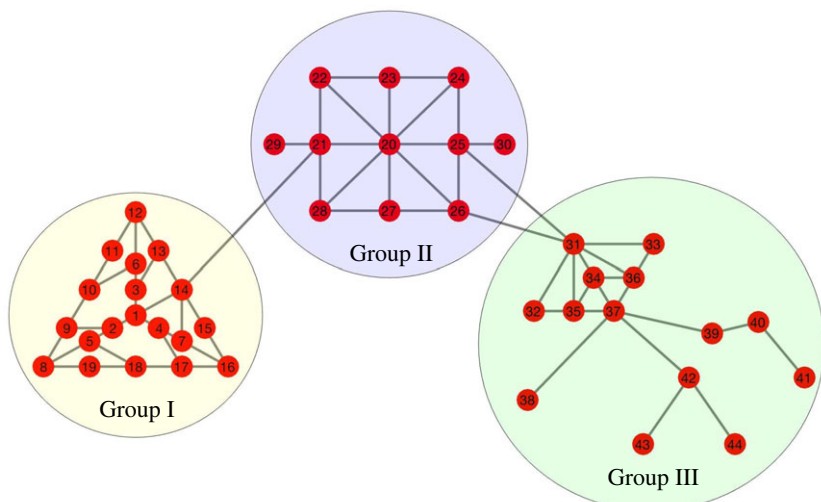

**Figure 6.** The three-group network. Group I (yellow) comprises nodes 1-19, Group II (blue) 21-30 and Group III (green) 31–44. Groups are sparsely connected.

**Table 2.** Numerical results for estimation of $\beta$, $\gamma$, $T$ and $\sigma/\alpha$ for the perturbed 20-barbell graph. Data are generated with an additional edge between nodes 2 and 40, which uses the specified fraction of the transition rate of other edges; parameter estimation uses the original model. See table 1 for description of values.

| transition rate | $\theta_\beta$ | $u_{\theta_\beta}$ (%) | $\theta_\gamma$ | $u_{\theta_\gamma}$ (%) | $\theta_T$ | $u_{\theta_T}$ (%) | $\sigma/\alpha$ | $u_{\sigma/\alpha}$ (%) |
|---|---|---|---|---|---|---|---|---|
| 10% | 1.0473 | 8.63 | 1.0228 | 7.87 | 0.9667 | 6.62 | 2.13 | 68.62 |
| 50% | 1.3283 | 14.58 | 1.2401 | 12.95 | 0.8008 | 12.67 | 3.51 | 64.86 |

results for this network use only observations of the infective populations (in contrast with results for the barbell graph, which additionally used observations of the recovered populations).

### 4.2.1. Parameter estimation

First, we attempt to recover $\beta$, $\gamma$ and $T$ (reference values 0.02, 0.3, 5, respectively) for a disease which starts at node 34 with $S_{34}(0) = 5$, $I_{34}(0) = 90$, $R_{34}(0) = 5$. All other nodes are fully susceptible, i.e. $S_i(0) = 100$, $I_i(0) = R_i(0) = 0$, for all $i \neq 34$.

Results are shown in figure 7a–c and in the first block of table 3. Sensors at the 7-node subset recover $\beta$, $\gamma$ and $T$ to within one standard deviation, while larger subsets recover all parameters with comparable

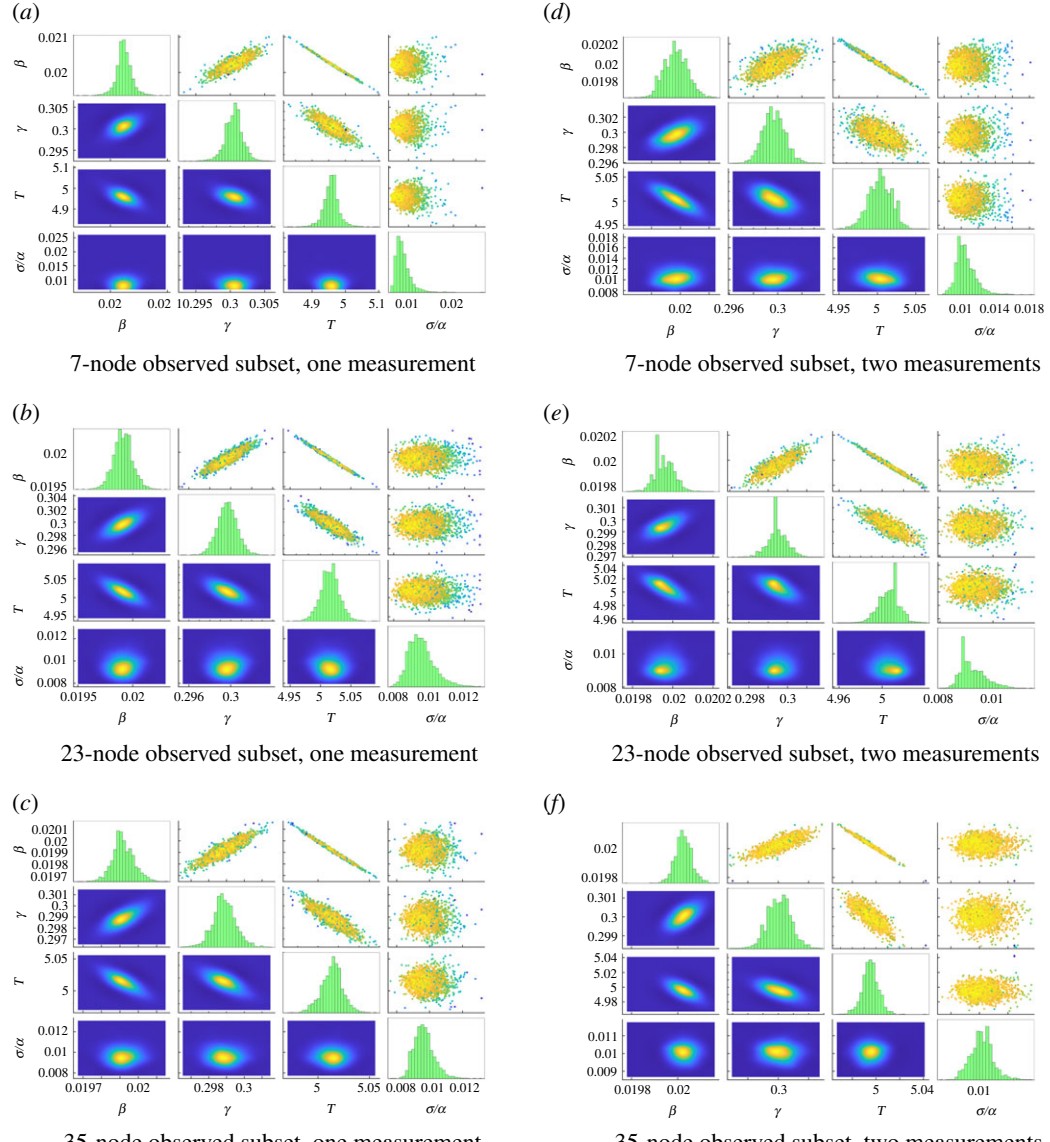

**Figure 7.** Parameter estimation results for the three-group network. (*a–c*) Parameter estimation for $\beta$, $\gamma$, $T$ based on information from 7-node, 23-node and 35-node subsets, respectively. Reference values are $\beta = 0.02$, $\gamma = 0.3$ and $T = 5$, with time step $\Delta t = 0.0005$. (*d–f*) Same experiment using two observations from each sensor separated by an interval $t = 1$. See figure 2 for description of subfigures. (*a*) Seven-node observed subset, one measurement, (*b*) 23-node observed subset, one measurement, (*c*) 5-node observed subset, one measurement, (*d*) 7-node observed subset, two measurements, (*e*) 23-node observed subset, two measurements, (*f*) 35-node observed subset, two measurements.

accuracy and greater precision: the 7-node 95% confidence interval for the scaled infection rate $\theta_\beta$ is [0.9986, 1.0253], which narrows to [0.9868, 1.0042] in the 23-node case. Increasing the number of observed nodes has the additional effect of accurately recovering the scaled noise level $\theta_{\sigma/\alpha}$, which is significantly underestimated when observing only the 7-node set.

Many real epidemic datasets include multiple observations over time; it is worth verifying that our approach can reasonably incorporate such observations. $T$ (time from infection to first sample) remains the only unknown time in this setting, as the relative timing between samples is known. Blocks 2–4 of table 3, the right column of figure 7 and the left column of figure 8 present results when considering additional samples taken at evenly spaced intervals of length 1. Including a second observation yields a significant reduction in estimated uncertainties, especially when observing only 7 nodes (uncertainties for $\beta$, $\gamma$, $T$ and $\sigma/\alpha$ are reduced by factors of 1.66, 1.38, 1.49 and 1.45, respectively). Comparatively, additional samples beyond the second yield diminishing returns (when observing the 7-node set, moving from two to four samples reduces uncertainties for $\beta$, $\gamma$, $T$ and $\sigma/\alpha$

**Table 3.** Numerical results for estimation of $\beta$, $\gamma$ and $T$ for the three-group network with $\sigma = 0.01\alpha$. Seven-node set is nodes 1–4, 20, 31 and 34; 23-node set is nodes 1–7, 20–28 and 31–37; and 35-node set is nodes 1–28 and 31–37. In the case of multiple samples (blocks 2–4), observations are evenly spaced in time with known interval 1. Perturbation experiment generated data with an additional edge between nodes 16 and 44. See table 1 for description of parameter values.

| experiment | observed set | $\theta_\beta$ | $u_\beta$ (%) | $\theta_\gamma$ | $u_\gamma$ (%) | $\theta_T$ | $u_T$ (%) | $\theta_{\sigma/\alpha}$ | $u_{\sigma/\alpha}$ (%) |
|---|---|---|---|---|---|---|---|---|---|
| 1 sample | 7-node | 1.0119 | 0.66 | 1.0017 | 0.42 | 0.9915 | 0.46 | 0.87 | 18.49 |
| | 23-node | 0.9955 | 0.44 | 0.9989 | 0.31 | 1.0032 | 0.31 | 0.96 | 7.72 |
| | 35-node | 0.9959 | 0.29 | 0.9963 | 0.22 | 1.0027 | 0.21 | 0.97 | 6.24 |
| 2 samples | 7-node | 0.9985 | 0.40 | 0.9990 | 0.31 | 1.0007 | 0.30 | 1.05 | 10.58 |
| | 23-node | 0.9979 | 0.25 | 0.9983 | 0.17 | 1.0014 | 0.19 | 0.92 | 5.08 |
| | 35-node | 1.0013 | 0.18 | 1.0001 | 0.13 | 0.9991 | 0.14 | 1.01 | 3.84 |
| 3 samples | 7-node | 1.0015 | 0.33 | 1.0014 | 0.23 | 0.9986 | 0.26 | 0.98 | 8.54 |
| | 23-node | 1.0011 | 0.19 | 1.0002 | 0.13 | 0.9990 | 0.16 | 0.99 | 3.99 |
| | 35-node | 1.0012 | 0.13 | 0.9996 | 0.09 | 0.9991 | 0.11 | 1.00 | 3.38 |
| 4 samples | 7-node | 1.0025 | 0.32 | 1.0011 | 0.17 | 0.9977 | 0.26 | 0.99 | 7.50 |
| | 23-node | 1.0004 | 0.16 | 1.0001 | 0.10 | 0.9995 | 0.13 | 1.01 | 3.37 |
| | 35-node | 0.9997 | 0.12 | 0.9993 | 0.08 | 1.0002 | 0.11 | 0.98 | 3.38 |
| perturbation | 35-node | 1.0019 | 2.51 | 0.9983 | 1.84 | 0.9996 | 1.77 | 6.93 | 6.52 |

by factors of 1.26, 1.80, 1.17 and 1.49, respectively). We additionally note that observing multiple samples over time can reduce the correlation between parameter uncertainties (e.g. the highly correlated posterior of $\beta$ and $\gamma$ in figure 7a when compared with the nearly uncorrelated ellipse of figure 8a).

As a basic example showing that the success of the approach is not reliant on this exact choice of epidemic model, we next consider a modified model wherein the infection rate of the disease varies by location. Specifically, we choose the infection rate to be the original $\beta_1 = 0.02$ within Group I, but to be doubled to $\beta_2 = 0.04$ in Groups II and III. As infectivity incorporates both characteristics of the disease itself and social behaviours (such as wearing masks or physically distancing), this represents a scenario wherein behaviours vary by community. Parameter estimation for this scenario using sensors at the 35-node subset appear in table 4. As in the original model, system parameters are recovered to good accuracy (all within 1%) and with low uncertainty (less than 1%), suggesting the efficacy of the approach is not limited to the particular model explored here.

As with the 20-barbell graph, we also consider model misspecification via perturbation of the network. The final row of table 3 presents parameter estimation results when observations at the 35-node subset are generated using an additional edge connecting nodes 16 and 44 (and thus Groups I and III); transition rates along the additional edge are chosen to match those of the edges connecting Groups II and III. As before, uncertainties for system parameter estimates ($\beta$, $\gamma$ and $T$) are larger than without the perturbation (by factors of 8.69, 8.53 and 8.54, respectively), though recovered parameter values are nonetheless accurate (scaled values off by less than 0.01), and so parameter estimation in this setting remains successful. The noise level $\sigma/\alpha$ is again overestimated, as it must additionally account for disagreement between the model used to generate the data and the model used to perform parameter estimation.

We next augment the parameter set $\underline{\theta}$ with the initial population vector $S_0$, $I_0$ and $R_0$ of the initially infected node, but take as known the observation time $T = 5$. In order that the reference values of all parameters be positive, the initial population vector at node 34 is altered to $S_{34}(0) = 5$, $I_{34}(0) = 90$, $R_{34}(0) = 5$. The scaled parameter set ($\theta_\beta$, $\theta_\gamma$, $\theta_{S_0}$, $\theta_{I_0}$, $\theta_{R_0}$, $\theta_{\sigma/\alpha}$) uses a uniform prior on $[0.02, 2] \times [0.02, 2] \times [0, 10] \times [0, 10] \times [0, 10] \times [0.001, 0.10]$.

Results are shown in figure 8d–f and appear numerically in table 5. Compared to estimation of $\beta$, $\gamma$ and $T$, correlations between parameters are generally weaker in this context, though there do exist clear relationships (e.g. larger $I_0$ necessitates smaller $\beta$ for the infection to spread at the same absolute rate). All sensor configurations recover the disease parameters $\beta$ and $\gamma$ accurately, with scaled values off by less than 0.01 in all cases. Conversely, uncertainty in the recovered distributions for $S_0$ and $R_0$ is

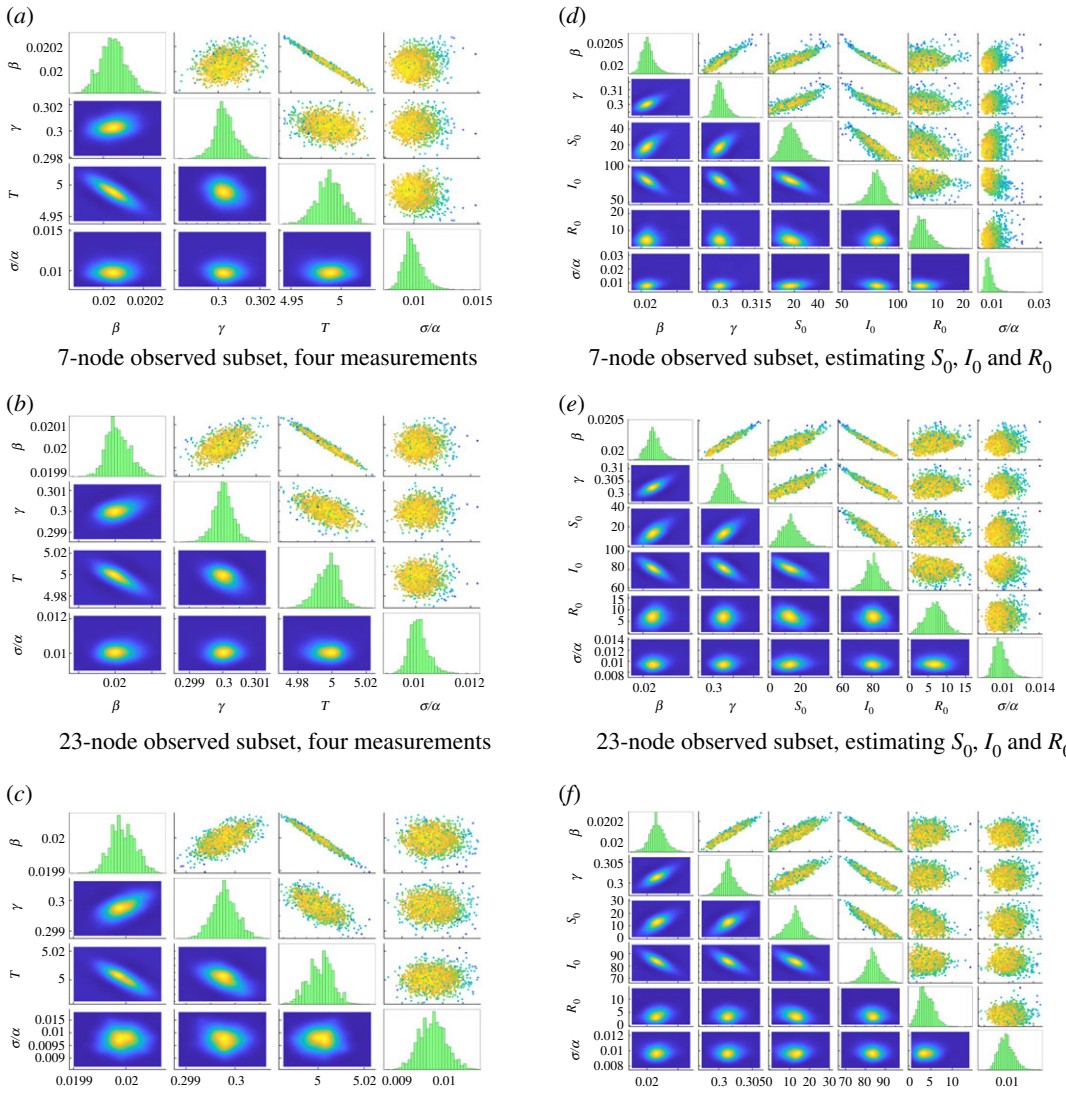

**Figure 8.** Additional parameter estimation results for the three-group network. ($a$–$c$) Estimating $\beta$, $\gamma$, $T$ using four measurements at intervals $t = 1$. Reference values are again $\beta = 0.02$, $\gamma = 0.3$ and $T = 5$, with time step $\Delta t = 0.0005$. ($d$–$f$) Estimating $\beta$, $\gamma$, $S_0$, $I_0$ and $R_0$ with one measurement at known time $T = 5$. Reference values are $\beta = 0.02$, $\gamma = 0.3$, $S_0 = 5$, $I_0 = 90$ and $R_0 = 5$, with time step $\Delta t = 0.02$. See figure 2 for description of subfigures. ($a$) Seven-node observed subset, four measurements, ($b$) 23-node observed subset, four measurements, ($c$) 35-node observed subset, four measurements, ($d$) 7-node observed subset, estimating $S_0$, $I_0$ and $R_0$, ($e$) 23-node observed subset, estimating $S_0$, $I_0$ and $R_0$, ($f$) 35-node observed subset, estimating $S_0$, $I_0$ and $R_0$.

**Table 4.** Numerical results for estimation of $\beta_1$, $\beta_2$, $\gamma$ and $T$ for the three-group network with $\sigma = 0.01\alpha$; and differing infection rates $\beta_1$ and $\beta_2$ between Group I and Groups II/III, respectively. Sensors were placed at the 35-node subset including nodes 1–28 and 31–37. See table 1 for description of parameter values.

| $\theta_{\beta_1}$ | $u_{\beta_1}$ (%) | $\theta_{\beta_2}$ | $u_{\beta_2}$ (%) | $\theta_\gamma$ | $u_\gamma$ (%) | $\theta_T$ | $u_T$ (%) | $\theta_{\sigma/\alpha}$ | $u_{\sigma/\alpha}$ (%) |
|---|---|---|---|---|---|---|---|---|---|
| 0.9926 | 0.97 | 1.0045 | 0.83 | 0.9924 | 0.66 | 1.0025 | 0.60 | 0.97 | 5.90 |

significantly higher than for other parameters, owing to the small ground-truth values for these parameters relative to the total population at the origin ($S_0 = R_0 = 5$ out of 100 individuals at time $t = 0$) and their correspondingly minor effect on the behaviour of the initial outbreak.

**Table 5.** Numerical results for estimation of $\beta$, $\gamma$, $S_0$, $I_0$ and $R_0$ for the three-group network with $\sigma = 0.01\alpha$. See table 1 for description of values.

| observed set | $\theta_\beta$ | $u_\beta$ (%) | $\theta_\gamma$ | $u_\gamma$ (%) | $\theta_{\sigma/\alpha}$ | $u_{\sigma/\alpha}$ (%) |
|---|---|---|---|---|---|---|
| 7-node | 1.0047 | 0.47 | 1.0022 | 0.87 | 0.86 | 24.72 |
| 23-node | 1.0056 | 0.38 | 1.0097 | 0.59 | 0.96 | 7.82 |
| 35-node | 1.0034 | 0.25 | 1.0048 | 0.39 | 0.98 | 6.26 |
| observed set | $\theta_{S_0}$ | $u_{S_0}$ (%) | $\theta_{I_0}$ | $u_{I_0}$ (%) | $\theta_{R_0}$ | $u_{R_0}$ (%) |
| 7-node | 3.5525 | 42.99 | 0.8874 | 8.56 | 1.0324 | 59.26 |
| 23-node | 2.7714 | 45.34 | 0.8969 | 7.02 | 1.3630 | 37.71 |
| 35-node | 2.4802 | 35.68 | 0.9291 | 4.77 | 0.7811 | 43.64 |

**Table 6.** Subset of model selection results for the three-group network. Recovered scaled parameters $\theta$ and uncertainties $u$ appear with the estimated log evidence for the model.

| model | log evidence | $Pr(M_j|D)$ | $\theta_\beta$ | $u_\beta$ (%) | $\theta_\gamma$ | $u_\gamma$ (%) | $\theta_{\sigma/\alpha}$ | $u_{\sigma/\alpha}$ (%) |
|---|---|---|---|---|---|---|---|---|
| $M_1$ | −45.6069 | 1.00 | 0.9996 | 0.06 | 0.9999 | 0.13 | 0.97 | 6.08 |
| $M_8$ | −298.8088 | ∼0.00 | 1.8226 | 2.92 | 1.0178 | 6.48 | 43.95 | 6.99 |
| $M_{14}$ | −289.9867 | ∼0.00 | 0.9474 | 2.11 | 0.9440 | 4.51 | 38.64 | 5.45 |
| $M_{21}$ | −319.5694 | ∼0.00 | 0.9160 | 4.59 | 0.9534 | 9.31 | 60.34 | 6.38 |
| $M_{32}$ | −342.3303 | ∼0.00 | 1.9701 | 1.66 | 0.7933 | 10.57 | 82.49 | 6.36 |
| $M_{43}$ | −358.6517 | ∼0.00 | 1.7586 | 10.41 | 1.4595 | 19.32 | 106.80 | 6.62 |

### 4.2.2. Origin of disease identification

Finally, we introduce a method for probabilistically identifying the origin of the epidemic (e.g. [30]) using the Bayesian model selection framework described in the Bayesian methodology section; recall that all observations are at the future time $T$, and so the origin may not be identified with certainty even when included in the set of observed nodes. We initialize the disease at node 1 with the standard initial configuration $S_1(0) = 5$, $I_1(0) = 95$ and $R_1(0) = 0$, with all other nodes fully susceptible with $S_i(0) = 100$, $I_i(0) = R_i(0) = 0$, and use corrupted observations of infective and recovered populations from the 35-node subset. Other parameter values are identical to those used in the previous section; in this case, $T = 5$ is assumed to be known, with $\beta$ and $\gamma$ estimated from observations. Defining the model class $M_j$ as the model under which the disease originated from node $j$ with the given initial vector, the log evidence for each model is generated from the model selection framework (see equation (3.4) in Bayesian methodology).

A representative selection of results appear in table 6. Model $M_1$, corresponding to the correct origin of the disease at node 1, has a significantly larger log evidence than all other models considered. Models which place the origin at increasingly distant points generate increasingly less accurate and more uncertain results; models $M_{32}$ and $M_{43}$, which originate the disease in Group III, find the estimated noise $\sigma$ to be two orders of magnitude larger than the reference value. Out of the models shown, if the correct model $M_1$ is not considered, the probabilities shift to 0.001 for $M_8$, 0.999 for $M_{14}$, and all other probabilities $\approx 10^{-13}$ or smaller, suggesting that topographic proximity to the true origin is the dominant factor in the evidence.

The effect of topographic proximity on the model evidence appears visually in figure 9. Models $M_i$, $i = 1, \ldots, 44$ correspond to the epidemic beginning at node $i$; each node in the graph is coloured by the estimated log evidence $\log Pr(M_i|\underline{D})$, where $\underline{D}$ are the noisy data obtained from a reference simulation beginning at node 1. In order that the fine detail be more visible, the colour mapping is non-uniform such that node 1 (log evidence −45.6) is the only node in its colour bin; other bins from log evidence −360 to −250 capture the range of behaviour in the remaining nodes. Evidence decays with topographic distance within the first group and becomes negligible in the second and third groups,

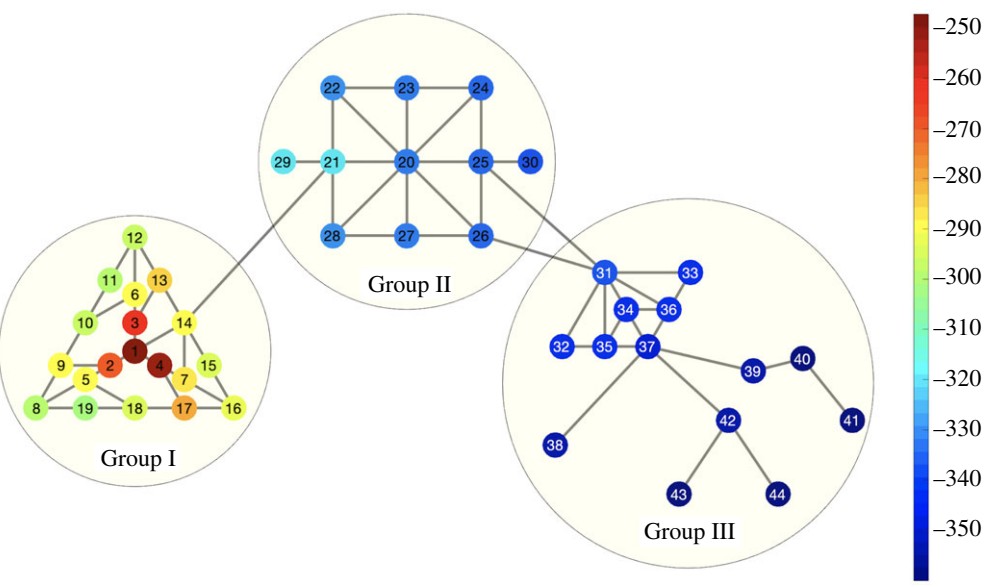

**Figure 9.** Origin identification model evidence by node, three-group network. Colour map of model selection log evidence for models initializing the epidemic at each node in the three-group network using noisy data from an epidemic simulation beginning at node 1. Non-uniform colour mapping (right label) emphasizes differences among incorrect models (log evidence −360 to −250); log evidence for the correct origin (node 1) is notably larger (log evidence −45.6).

**Table 7.** Subset of model selection results for the perturbed three-group network. Data are generated with an additional edge between nodes 16 and 44; parameter estimation uses the original model. Recovered scaled parameters $\theta$ and uncertainties $u$ appear with the estimated log evidence for the model.

| model | log evidence | $Pr(M_j\|D)$ | $\theta_\beta$ | $u_\beta$ (%) | $\theta_\gamma$ | $u_\gamma$ (%) | $\theta_{\sigma/\alpha}$ | $u_{\sigma/\alpha}$ (%) |
|-------|--------------|--------------|----------------|---------------|-----------------|----------------|--------------------------|-------------------------|
| $M_1$ | −90.8875 | 1.00 | 0.9983 | 0.13 | 0.9941 | 0.28 | 1.97 | 6.60 |
| $M_8$ | −298.6711 | ∼0 | 1.8138 | 3.10 | 1.0184 | 7.16 | 44.38 | 6.90 |
| $M_{14}$ | −289.9163 | ∼0 | 0.9514 | 1.87 | 0.9375 | 4.02 | 38.12 | 4.76 |
| $M_{21}$ | −319.2385 | ∼0 | 0.9160 | 4.15 | 0.9492 | 9.04 | 60.19 | 5.98 |
| $M_{32}$ | −342.0946 | ∼0 | 1.9786 | 1.40 | 0.7803 | 9.49 | 81.80 | 5.85 |
| $M_{43}$ | −358.7536 | ∼0 | 1.7745 | 10.82 | 1.4404 | 19.58 | 107.39 | 6.93 |

highlighting the improbability of the epidemic starting at these distant nodes and producing the noisy observations $\underline{D}$ corresponding to an epidemic outbreak at node 1.

Lastly, we attempt origin identification via the same model selection approach for the perturbed model which generates data using an additional edge connecting nodes 16 and 44; results appear in table 7. Despite the misspecification, the correct model $M_1$ is selected with near certainty, though its log evidence is many orders of magnitude smaller than in the no-perturbation experiment (−90.8875 versus −45.6069) owing to the high level of noise required to explain disagreements with the observed data. Other models largely have similar evidence and recovered parameter values as before, suggesting this type of misspecification to be insignificant compared to the overwhelming unlikelihood of observing significant early spread centred around a different node.

## 5. Discussion

We found that Bayesian UQ via TMCMC effectively recovered SIR network model parameters, such as the infection rate $\beta$ and recovery rate $\gamma$, using only noisy observations from a limited set of nodes. The approach was tested on two example networks with distinct topologies, with two possible sets of observed data (infective populations versus both infective and recovered populations), using different sets of free parameters, and in a number of additional contexts (in particular, with model

perturbations and with multiple observations at each node). Given its explicit estimation of parameter uncertainty, the framework permits comparison of precision in distinct scenarios—for example, uncertainties in recovered parameter values were found to be inversely related to the number of sensors (for the three-group network, increasing the number of sensors from 7 to 23 decreased parameter uncertainties by a factor of 1.2–1.8).

The 20-barbell graph, wherein only pairs of nodes were observed, provided additional insight into the effect of sensor placement on the uncertainty of parameter estimates. Sensors which were close together, e.g. nodes 3 and 12 (whose connectivity is identical), yielded similar noisy data, thereby affording less information about the underlying dynamics. By contrast, placing sensors on opposite sides of the network to gain information about dynamics on different timescales yielded significantly less uncertainty. Given limited resources for monitoring an epidemic, it may thus be beneficial to track a set of communities which are at varying stages of outbreak rather than allocating resources directly to those communities nearest to which the disease was initially observed.

Our approach proved robust both to perturbations in the model and to increased observational noise. Results for the 20-barbell graph with increased noise level $\sigma = 0.05\alpha$, five times the original $\sigma = 0.01\alpha$, recovered reference parameters with comparable accuracy. For the three-group network, we were also able to identify the disease origin with near certainty via selection among models corresponding to potential starting points, even when data were generated from a distinct model with an additional edge; model selection thus has the potential to locate real outbreaks even when observations begin well after the time of infection and the network structure is not known exactly [30–32]. We remark that the evidence calculation required for this procedure is an intermediate step of the Bayesian UQ framework, and so it does not incur any additional computational cost.

Broadly speaking, our results suggest that parallel implementations of Bayesian UQ in frameworks such as $\Pi$4U have great potential to perform statistical inference in real-world noisy settings, even when the underlying mathematical models have significant complexity and inherent modelling error. The Bayesian framework accurately and efficiently recovered system parameters for our network epidemic model, providing an approach for robust epidemic modelling and tracking in a rigorous probabilistic setting which can be further refined and tested by leveraging more complex population models (e.g. recent human mobility models [39–41]), observation models (e.g. partial observations [42–44]) and real datasets. The present framework can be readily deployed in conjunction with a wide range of computational models, and so we believe it will have broad practical relevance for the future prediction and management of epidemics.

Data accessibility. Data and relevant code for this research work are stored in GitHub: https://github.com/cselab/pi4u and have been archived within the Zenodo repository: https://doi.org/10.5281/zenodo.4015102.

Authors' contributions. K.L., G.A., C.B., Z.C. and P.H. developed code, ran simulations, visualized and analysed data and drafted the manuscript; C.P., P.K. and A.M. edited the manuscript and conceived, designed and coordinated the study. All authors gave final approval for publication.

Competing interests. We declare we have no competing interests.

Funding. K.L. and A.M. were partially supported by the NSF through grant nos. DMS-1521266 and DMS-1552903. C.P. was supported by the European Union's Horizon 2020 research and innovation programme under the Marie Sklodowska-Curie grant agreement no. 764547. P.K. was supported by the European Research Council Advanced Investigator Award (grant no. 34117).

Acknowledgements. Parts of this research were conducted using computational resources and services at the Center for Computation and Visualization, Brown University. Computer time has also been generously provided by the Swiss Center for Supercomputing (CSCS).

# Appendix A

## A.1. Further details on transitional Markov chain Monte Carlo

### A.1.1. Numerical error

In this section, we describe the application of TMCMC to a simple example (sampling from a known distribution) where the error can be easily characterized. Further details on this analysis can be found in [45].

We first consider a $d$-dimensional multivariate Gaussian distribution with mean $\mu = 0$ and a random covariance matrix $\Sigma$. We then use TMCMC to estimate the parameters $\mu$ and $\Sigma$, denoting the estimated

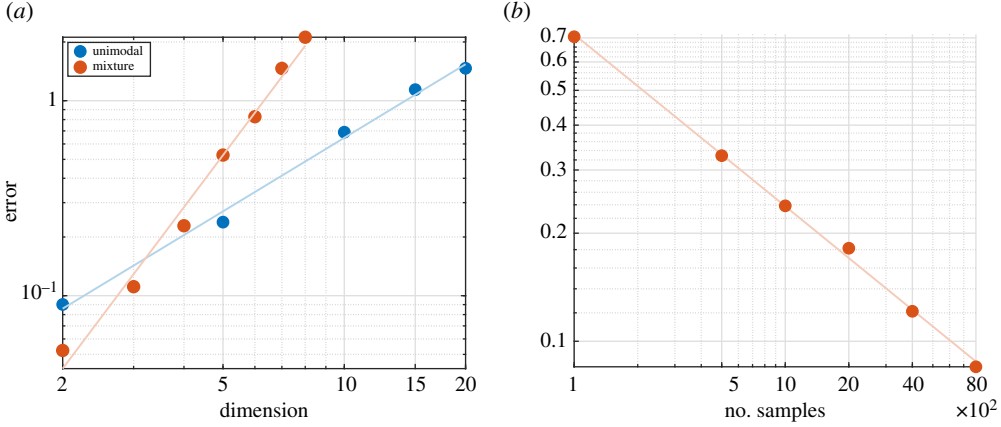

**Figure 10.** Numerical error estimates for multivariate Gaussian sampling. (*a*) Logarithmic plot of average error for a multivariate Gaussian distribution (blue) and a mixture of two Gaussians (red) as a function of dimension; linear fits are shown with solid lines. (*b*) Logarithmic plot of average error for a five-dimensional Gaussian distribution with respect to number of samples.

values as $\overline{\mu}$ and $\overline{\Sigma}$, respectively. Numerical error is calculated as

$$e = \frac{1}{2}\left(\frac{1}{d}\sum_{i=1}^{d}|\bar{\mu}_i - \mu_i|\right) + \frac{1}{2}\left(\frac{1}{d^2}\sum_{i,j=1}^{d}|\bar{\Sigma}_{i,j} - \Sigma_{i,j}|\right), \tag{A 1}$$

i.e. the equal average of the error in estimating $\mu$ and the error in estimating $\Sigma$, where each term uses the average $L_1$ error across components. We use the scaling parameter $b = 0.2$ (see equation (3.8)) and a uniform prior on $[-10, 10]^d$. Each case is repeated 100 times and the errors averaged to obtain a final estimate for the numerical error.

The blue points in figure 10*a* show the results obtained using $N_0 = 1000$ samples for a range of dimensions $d = 2, 5, 10, 15, 20$; error unsurprisingly increases with dimensionality. Figure 10*b* shows the results for fixing the dimensionality $d = 5$ and instead varying the number of samples $N_0$, showing error to decrease with samples. A linear fit of error versus samples yields a slope of $-0.5$, i.e. a convergence rate of $1/\sqrt{N_0}$.

As a comparison, we also compute results for a mixture of two Gaussians with means at $\mu_1 = (-5, \ldots, -5)$ and $\mu_2 = -\mu_1$ and equal covariances randomly generated as before. To estimate parameters in this setting, we divide samples into two groups with a simple clustering algorithm based on Euclidean distance. The red points in figure 10*a* show the average numerical error using $N_0 = 5000$ samples as a function of dimension $d$ from 2 to 8. In comparison to the previous case, the error is lower in low dimensions ($d = 2, 3$) but scales worse into higher dimensions.

### A.1.2. Comparison of TMCMC with nested sampling

In this section, we compare a selection of TMCMC results from the main text to the results which would be obtained using nested sampling, an alternative sampling method which also provides Bayesian estimates of model evidence [46]. (Most sampling methods do not estimate model evidence, and would thus make for an apples-to-oranges comparison.) We use the model and parameters corresponding to figure 2*b*, the 20-barbell graph with one sensor on each side and a sampling time $T = 5$.

Figure 11*a*–*c* shows the results of parameter estimation in this scenario using TMCMC with 10 000, 15 000 and 20 000 samples, respectively. Recall from the previous section that TMCMC converges with sample size as $1/\sqrt{N_0}$, i.e. more samples yields a lower error. The similarity between the three figures suggests that TMCMC has already converged to visually indistinguishable accuracy by this number of samples.

As an aside, we note that the convergence of TMCMC also varies with the number of intermediate generations, which is controlled by a hyperparameter of the method; similar studies have found comparable results using a wide range of values for this parameter, including the values used in this paper.

Figure 11*d* shows results using the nested sampling approach with 1000 live samples and 8500 final samplings. Joint and marginal distributions are indiscernable from the results of TMCMC, suggesting that both methods were able to accurately sample from this posterior. We conclude that the other

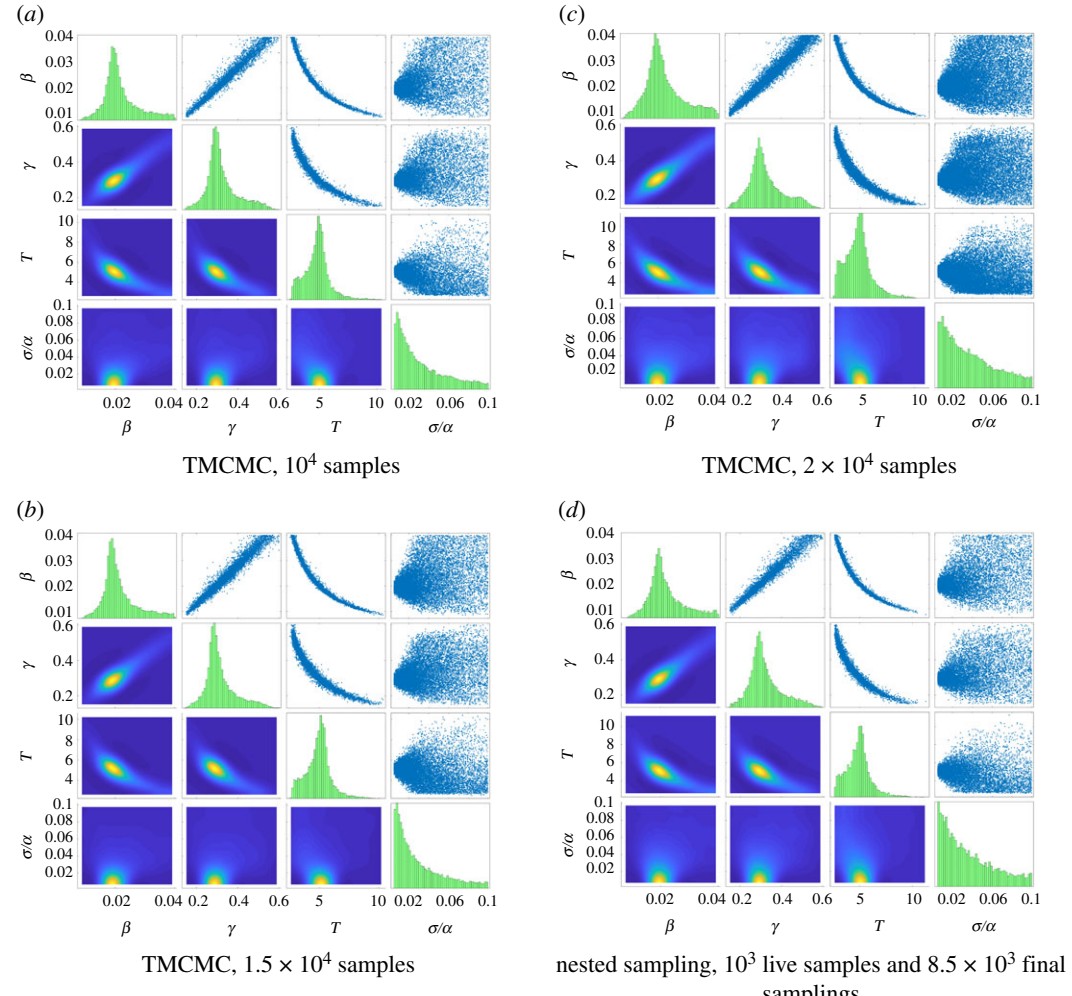

**Figure 11.** TMCMC and nested sampling results for the posterior distribution of figure 2b. (a) TMCMC, $10^4$ samples, (b) TMCMC, $1.5 \times 10^4$ samples, (c) TMCMC, $2 \times 10^4$ samples, (d) nested sampling, $10^3$ live samples and $8.5 \times 10^3$ final samplings.

advantages of TMCMC, most notably its efficient parallel implementation, make it an excellent choice for this sort of computationally intensive model selection.

### A.1.3. Convergence by generation

Figure 12 shows six intermediate generations of TMCMC convergence for the scenario of figure 2b, the same scenario used to draw a comparison with nested sampling above. These intermediate generations illustrate well the gradual transition from the uniform prior (Generation 0) to the sharply peaked posterior (Generation 10).

### A.1.4. High-performance implementation

$\Pi$4U [28] is a platform-agnostic task-based framework for UQ that supports nested parallelism and automatic load balancing in large-scale computing architectures. The software is open-source and includes HPC implementations for both multi-core and GPU clusters of algorithms such as transitional Markov chain Monte Carlo [27,38] and approximate Bayesian computational subset-simulation [47]. The irregular, dynamic and multi-level task-based parallelism of the algorithms (figure 13a) is expressed and fully exploited by means of the TORC runtime library [27]. TORC is a software library for programming and running unaltered task-parallel programs on both shared and distributed memory platforms. TORC orchestrates the scheduling of function evaluations on the cluster nodes (figure 13b). The parallel framework includes multiple features, most prominently its inherent load balancing, fault-tolerance and high reusability. As a specific example, the TMCMC

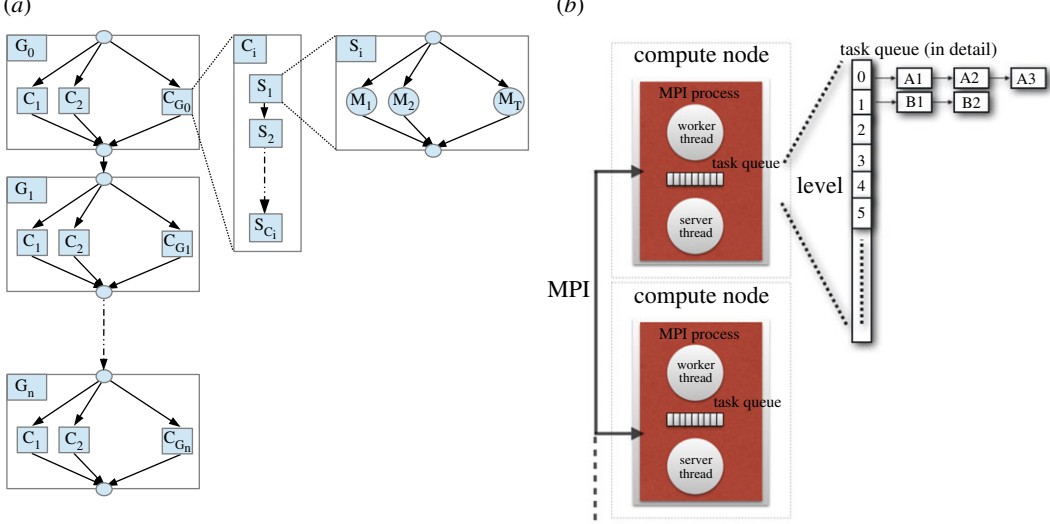

**Figure 12.** Samples from six intermediate generations of TMCMC for the scenario of figure 2*b*.

**Figure 13.** Task graph of the TMCMC algorithm (*a*) and parallel architecture of the TORC library (*b*).

method within $\Pi$4U is able to achieve an overall parallel efficiency of more than 90% on 1024 compute nodes of Swiss supercomputer Piz Daint running hybrid MPI+GPU molecular simulation codes with highly variable time-to-solution between simulations with different interaction parameters.

## A.2. Transition matrices for three-group network

Transition rates were chosen to vary among groups in the three-group network in order to test the robustness of our approach to non-uniform rates. Populations moved between Group I and Group II (via the edge connecting node 14 to node 21) at a rate of 0.4, while the transition rate between Group II and Group III (via the edges connecting node 31 to nodes 25 and 26) was 0.2. Rates ($\lambda_{i,j}$, $\eta_{i,j}$, $g_{i,j}$) within Group I were selected randomly to be either (0, 2, 0.05, 0.1) or (0.1, 0.1, 0.2), those for Group II were selected randomly to be either (0.15, 0.2, 0.1) or (0.3, 0.1, 0.1) and Group III had uniform rates of 0.05.

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
