## [Reviewer comments · Royal Society Open Science]

Review History

RSOS-200531.R0 (Original submission)

Review form: Reviewer 1

Is the manuscript scientifically sound in its present form?

No

Are the interpretations and conclusions justified by the results?

Yes

Is the language acceptable?

Yes

Do you have any ethical concerns with this paper?

No

Have you any concerns about statistical analyses in this paper?

Yes

Recommendation?

Major revision is needed (please make suggestions in comments)

Comments to the Author(s)

See attachment (Appendix A).

Decision letter (RSOS-200531.R0)

Dear Dr Matzavinos,

The editors assigned to your paper ("Data-driven prediction and origin identification of epidemics in population networks") have now received comments from reviewers. We would like you to revise your paper in accordance with the referee and Associate Editor suggestions which can be found below (not including confidential reports to the Editor). Please note this decision does not guarantee eventual acceptance.

Please submit a copy of your revised paper before 23-Jul-2020. Please note that the revision deadline will expire at 00.00am on this date. If we do not hear from you within this time then it will be assumed that the paper has been withdrawn. In exceptional circumstances, extensions may be possible if agreed with the Editorial Office in advance. We do not allow multiple rounds of revision so we urge you to make every effort to fully address all of the comments at this stage. If deemed necessary by the Editors, your manuscript will be sent back to one or more of the original reviewers for assessment. If the original reviewers are not available, we may invite new reviewers.

- Data accessibility

<http://datadryad.org/submit?journalID=RSOS&manu=RSOS-200531>

- Competing interests

- Authors' contributions

- Acknowledgements

- Funding statement

on behalf of Professor Tim Rogers (Associate Editor) and Mark Chaplain (Subject Editor)
openscience@royalsociety.org

Comments to Author:

Reviewers' Comments to Author:

Reviewer: 1

Comments to the Author(s)

See attachment. (Review_RSOS_200531.pdf)

Author's Response to Decision Letter for (RSOS-200531.R0)

See Appendix B.

RSOS-200531.R1 (Revision)

Review form: Reviewer 1

Is the manuscript scientifically sound in its present form?

Yes

Are the interpretations and conclusions justified by the results?

Yes

Is the language acceptable?

Yes

Do you have any ethical concerns with this paper?

No

Have you any concerns about statistical analyses in this paper?

No

Recommendation?

Accept with minor revision (please list in comments)

Comments to the Author(s)

Thanks for your effort to address my comments. I would like to raise the following minor issues.

- Regarding the choice of Gaussian errors:

The correct statement is that the Gaussian distribution maximizes entropy over probability distributions with support the real line and specified mean and variance. However, apart from the information theoretic argument, the Gaussian distribution assumes unimodality, symmetricity, and no kurtosis. I think it is misleading to present it as an agnostic choice. I suggest rephrasing this.

- In Fig 2 D and F, it seems that the distribution of the parameters is bimodal. Can you comment on this and explain? What the two modes correspond to, why is this observed and why in this instance and not the others of the same figure?
- Regarding the paragraph "Numerical error" in Appendix:
If this basic analysis is already done in the referenced paper then there is nothing new here and a reference to the paper would suffice. The relevant question is what are the implications of this problem in the high-dimensional model considered in the manuscript. I suggest replacing this section with a paragraph addressing the later issue.

Decision letter (RSOS-200531.R1)

Dear Dr Matzavinos

On behalf of the Editors, we are pleased to inform you that your Manuscript RSOS-200531.R1 "Data-driven prediction and origin identification of epidemics in population networks" has been accepted for publication in Royal Society Open Science subject to minor revision in accordance with the referees' reports. Please find the referees' comments along with any feedback from the Editors below my signature.

Please submit your revised manuscript and required files (see below) no later than 7 days from today's (ie 04-Dec-2020) date. Note: the ScholarOne system will 'lock' if submission of the revision is attempted 7 or more days after the deadline. If you do not think you will be able to meet this deadline please contact the editorial office immediately.

on behalf of Professor Tim Rogers (Associate Editor) and Mark Chaplain (Subject Editor)
 openscience@royalsociety.org

Reviewer comments to Author:

Reviewer: 1

Comments to the Author(s)

Thanks for your effort to address my comments. I would like to raise the following minor issues.

- Regarding the choice of Gaussian errors:

The correct statement is that the Gaussian distribution maximizes entropy over probability distributions with support the real line and specified mean and variance. However, apart from the information theoretic argument, the Gaussian distribution assumes unimodality, symmetricity, and no kurtosis. I think it is misleading to present it as an agnostic choice. I suggest rephrasing this.

- In Fig 2 D and F, it seems that the distribution of the parameters is bimodal. Can you comment on this and explain? What the two modes correspond to, why is this observed and why in this instance and not the others of the same figure?

- Regarding the paragraph "Numerical error" in Appendix:

If this basic analysis is already done in the referenced paper then there is nothing new here and a reference to the paper would suffice. The relevant question is what are the implications of this problem in the high-dimensional model considered in the manuscript. I suggest replacing this section with a paragraph addressing the later issue.

===PREPARING YOUR MANUSCRIPT===

If you have been asked to revise the written English in your submission as a condition of publication, you must do so, and you are expected to provide evidence that you have received language editing support. The journal would prefer that you use a professional language editing service and provide a certificate of editing, but a signed letter from a colleague who is a native

speaker of English is acceptable. Note the journal has arranged a number of discounts for authors using professional language editing services (<https://royalsociety.org/journals/authors/benefits/language-editing/>).

===PREPARING YOUR REVISION IN SCHOLARONE===

-- If you have uploaded ESM files, please ensure you follow the guidance at <https://royalsociety.org/journals/authors/author-guidelines/#supplementary-material> to include a suitable title and informative caption. An example of appropriate titling and captioning

may be found at https://figshare.com/articles/Table_S2_from_Is_there_a_trade-off_between_peak_performance_and_performance_breadth_across_temperatures_for_aerobic_sc_ope_in_teleost_fishes_/3843624.

Author's Response to Decision Letter for (RSOS-200531.R1)

See Appendix C.

Decision letter (RSOS-200531.R2)

This year has been very difficult for everyone, and we want to take the opportunity to thank you for your continued support in 2020.

The Royal Society Open Science editorial office will be closed from the evening of Friday 18 December 2020 until Monday 4 January 2021. We will not be responding during this time. If you have received a deadline within this time period, please contact us as soon as possible to allow us to extend the deadline. If you receive any automated messages during this time asking you to meet a deadline, we offer apologies and invite you to respond after the festive period or during normal working hours.

With our best for a peaceful festive period and New Year, and we look forward to working with you in 2021.

Dear Dr Matzavinos,

It is a pleasure to accept your manuscript entitled "Data-driven prediction and origin identification of epidemics in population networks" in its current form for publication in Royal Society Open Science.

on behalf of Professor Tim Rogers (Associate Editor) and Mark Chaplain (Subject Editor)
openscience@royalsociety.org

Appendix A

The paper discusses a framework for Bayesian model selection and parameter estimation of a dynamical system motivated by epidemiology. More specifically, a Transitional MCMC algorithm is used to infer the parameters of a network SIR model. The same algorithm is used to compare different versions of the network SIR model, e.g. different initial states of the system. The authors use simulated data from two different networks to test their method under a variety of parameter settings.

The subject is timely and important and the manuscript well written. The approach appears to be sound and the results interesting. It clearly demonstrates potential for an excellent publication.

However, I find the current version of the manuscript very limited in terms of description and analysis of their approach and especially in terms of their results. In the following I make some specific comments along these lines. I split my comments into major and minor issues to help the authors understand better my concerns.

Major:

1. To understand why the TMCMC method is useful, we need to have a comparison with other methods. I do not want to restrict the authors by suggesting specific targets but some comparison with more basic MCMC algorithms, algorithms similar to TMCMC, and/or state-of-the-art methods would benefit the paper substantially.
2. The convergence of the chains to posterior distributions is not discussed. There are no Markov Chains plotted, nor any convergence test results reported. An analysis of how the parameters of the TMCMC algorithm (e.g. number of transitions) affect convergence but also computational speed is necessary.
3. The sensitivity analysis performed is very limited at the moment. Various sensitivity questions, e.g. variability levels, priors, the network structure, the observed time-point (location and number), need to be validated further. In terms of network structure, the authors considered some simulated data where the network structure is not correct, but it seems to me that this is still very close to truth. One could imagine many scenarios, e.g. what if the connections in the graph are seriously underestimated, what if infection rate constants are different in different communities, but I prefer to leave the choice to the authors.
4. The scalability of the method is not discussed. I couldn't see any comments related to the computational speed and how it compares with other methods (see point 1). Have the authors tested the method in much bigger networks? Have they tested the limits of the method in terms of network structure? What are the critical parameters that affect scalability? How the speed is affected when more observations are available?

Minor:

1. The properties of the TMCMC need to be discussed. For example, are there any analytical/asymptotic results on the convergence of the simulated distributions to the posterior distribution?
2. I suggest creating a diagram of the epidemic model close to eq (4) to help the reader.

3. First paragraph in section “Bayesian Methodology”: it is worth mentioning another important source of noise which is due to that transition times, e.g. I to R , in reality are themselves noisy (e.g. dependent on the subject, current state) something often referred as “intrinsic stochasticity”.
4. Line 14, page 6: comment on the choice of uniform prior on models. A Bayesian approach would naturally assume that there is some prior information on the choice of the model.
5. The normality assumption for the errors is a strong assumption, especially given that so many noise components are assumed to be only described by a single parameter. I suggest a comment on this.
6. Line 14, p. 6: better avoid j index as already used.
7. At the beginning of the results section, I suggest emphasizing that you discuss simulated data, e.g. by a title.
8. Figure 2: The lower diagonal plots does not seem to match upper diagonal. Do they? Are they supposed to match? Please clarify what data are used for lower diagonal.
9. Line 20 p.14: Up to that point I thought T is assumed a known parameter. Please clarify.
10. Fig. 6: It's hard to read the numbers in the nodes.
11. Line 24, p.19: How and why does the algorithm distinguish between $M1$ and nearby nodes $M2$, $M3$ or $M4$?
12. Line 30, p. 19: Similarly to point 11, why $M8$ and $M14$ and not $M2$, $M3$ and $M4$?
13. The second and third paragraph of Discussion are a repetition of earlier results. I a more useful discussion could be possible improvements of the method and comparisons with other methods.

Appendix B

Response to referee's comments on:
Data-driven prediction and origin identification of epidemics
in population networks

Manuscript ID: RSOS-200531 R1

The authors are grateful to the reviewer for a thorough and constructive set of comments. While addressing these comments, we discovered that the code used to set up the networks in the previous version of the manuscript had a minor bug and was consistently omitting an edge in the network structure. This is now corrected, resulting, for instance, in a slightly modified Fig. 3. We are happy to re-submit the accompanying revised manuscript which we feel has addressed all of the reviewer's concerns.

Major issues:

To understand why the TMCMC method is useful, we need to have a comparison with other methods. I do not want to restrict the authors by suggesting specific targets but some comparison with more basic MCMC algorithms, algorithms similar to TMCMC, and/or state-of-the-art methods would benefit the paper substantially.

We now compare the performance of the TMCMC method on the barbell network to that of the nested sampling algorithm. The latter is one of the main competitors to TMCMC for model selection, as both approaches provide efficient estimates of the Bayesian model evidence in addition to sampling the posterior distribution. These new results can be found in the appendix.

The convergence of the chains to posterior distributions is not discussed. There are no Markov Chains plotted, nor any convergence test results reported. An analysis of how the parameters of the TMCMC algorithm (e.g. number of transitions) affect convergence but also computational speed is necessary.

A thorough computational analysis of the rate of convergence of TMCMC is now provided in the appendix, where errors in the parameter estimates are quantified as functions of the dimensionality of the parameter space and the number of samples (see Figures 10A and 10B). We also present and discuss visualizations of the intermediate distributions generated by TMCMC (see Figure 12).

The sensitivity analysis performed is very limited at the moment... One could imagine many scenarios, e.g. what if the connections in the graph are seriously underestimated, what if infection rate constants are different in different communities, but I prefer to leave the choice to the authors.

In addition to the edge deletion/addition experiments discussed in the manuscript, we now perform parameter estimation experiments with differential infectivity rates, as requested by the reviewer.

The scalability of the method is not discussed. I couldn't see any comments related to the computational speed and how it compares with other methods (see point 1). Have the authors tested the method in much bigger networks? Have they tested the limits of the method in terms of network structure? What are the critical parameters that affect scalability? How the speed is affected when more observations are available?

These questions relate to the convergence rate of the TMCMC method and are now addressed in the appendix.

Minor issues:

The properties of the TMCMC need to be discussed. For example, are there any analytical/asymptotic results on the convergence of the simulated distributions to the posterior distribution?

The convergence of TMCMC is readily reduced to the ergodicity of the Metropolis algorithm. The rate of convergence is a highly non-trivial issue, and we have now included a thorough timing study in the appendix.

I suggest creating a diagram of the epidemic model close to eq (4) to help the reader.

We feel that the arrow schematic on page 3 conveys all the important information discussed in the text.

First paragraph in section "Bayesian Methodology": it is worth mentioning another important source of noise which is due to that transition times, e.g. I to R, in reality are themselves noisy (e.g. dependent on the subject, current state) something often referred as "intrinsic stochasticity".

We have now included explicit mention of intrinsic stochasticity as an example of model error.

Line 14, page 6: comment on the choice of uniform prior on models. A Bayesian approach would naturally assume that there is some prior information on the choice of the model.

The uniform distribution is a maximum entropy distribution over a bounded domain, i.e., it represents the most "agnostic" choice for bounded domains. We have now added a comment clarifying this, along with a reference to the literature.

The normality assumption for the errors is a strong assumption, especially given that so many noise components are assumed to be only described by a single parameter. I suggest a comment on this.

The Gaussian distribution is a maximum entropy distribution over the real line (unbounded domain), i.e., it represents the most "agnostic" choice. We have now added a comment clarifying this, along with a reference to the literature.

Line 14, p. 6: better avoid j index as already used.

Please note that the usage of index j is correct. The *while* loop in the pseudocode advances the index j in $\theta_{j,k}$.

At the beginning of the results section, I suggest emphasizing that you discuss simulated data, e.g. by a title.

We now explicitly mention the use of simulated data in the first sentence of the Results section. We note that the use of simulated data was already emphasized in the second paragraph of the same section on page 7, which is the beginning of the Results section.

Figure 2: The lower diagonal plots does not seem to match upper diagonal. Do they? Are they supposed to match? Please clarify what data are used for lower diagonal.

The lower diagonal plots are obtained by interpolating and transposing the upper diagonal ones. We have now added a comment clarifying this.

Line 20 p.14: Up to that point I thought T is assumed a known parameter. Please clarify.

Since all infection dynamics are initialized at time $t = 0$, assuming T is known is equivalent to assuming knowledge of the time of the first transmission event. Through most of the paper T is *not* assumed known, which allows us to estimate when the disease first appeared by estimating T as the time elapsed after the first transmission event. Only the augmented parameter set (Table 5) and the model selection results (Tables 6, 7) assume T to be known.

Fig. 6: It's hard to read the numbers in the nodes

It is possible that the generated manuscript was downsampled. All our figures are high quality and we are confident that the professionals at the Royal Society will confirm the image quality in the final product.

Line 24, p.19: How and why does the algorithm distinguish between M_1 and nearby nodes M_2 , M_3 or M_4 ?

The SIR model produces different dynamics when initialized in these nearby nodes, yielding differing levels of agreement with the noisy observed data. We note that although Table 6 displays only a subset of model selection results, it purposefully includes node 14, which is directly adjacent to node 1, in order to show the ability of the approach to distinguish between nearby nodes. The evidence supporting M_{14} is orders of magnitude lower than the accurate model M_1 .

Line 30, p. 19: Similarly to point 11, why M_8 and M_{14} and not M_2 , M_3 and M_4 ?

See answer above; the table shows only a representative selection of results. The full results are visualized in Figure 9, which states that the log evidence for all incorrect models (including M_2 , M_3 , and M_4) is less than -250, many times smaller than the -45.6 observed for M_1 .

The second and third paragraph of Discussion are a repetition of earlier results. I a more useful discussion could be possible improvements of the method and comparisons with other methods.

We have now added an extensive appendix with a timing study of our TCMC implementation and comparisons with other methods. We feel that a concluding synopsis of our results in the Discussion section improves the readability of the manuscript.

Appendix C

Response to referee's comments on:
Data-driven prediction and origin identification of epidemics
in population networks

Manuscript ID: RSOS-200531 R2

The authors would like to thank the reviewer for his/her constructive comments and suggestions. We are happy to resubmit the accompanying revised manuscript which we feel has addressed all of the reviewer's concerns.

The correct statement is that the Gaussian distribution maximizes entropy over probability distributions with support the real line and specified mean and variance. However, apart from the information theoretic argument, the Gaussian distribution assumes unimodality, symmetricity, and no kurtosis. I think it is misleading to present it as an agnostic choice. I suggest rephrasing this.

Line 22, page 5: We have now replaced the sentence "...the Gaussian distribution, which maximizes entropy over the real numbers, is the most agnostic choice of error" with "...the multivariate normal distribution maximizes entropy over the class of probability distributions on \mathbb{R}^m with specified mean and covariance matrix."

In Fig 2 D and F, it seems that the distribution of the parameters is bimodal. Can you comment on this and explain? What the two modes correspond to, why is this observed and why in this instance and not the others of the same figure?

Figures 2D and 2F represent estimates associated with sensor configurations that sample only one of the two communities at a relatively early stage. As discussed on page 8, it is expected that these sensor configurations will provide less accurate estimates as compared to those in Figure 2E, where samples from both communities have been utilized.

Regarding the paragraph "Numerical error" in Appendix: If this basic analysis is already done in the referenced paper then there is nothing new here and a reference to the paper would suffice. The relevant question is what are the implications of this problem in the high-dimensional model considered in the manuscript. I suggest replacing this section with a paragraph addressing the later issue.

We believe that the appendix in its current form facilitates the readability of the manuscript and have chosen to retain the numerical error section. Regarding the implications for the epidemic model considered in the manuscript, please see the comparison of the performance of TMCMC to that of nesting sampling (page 22). This is specific to the model and parameters corresponding to Figure 2B.